# Climate change over the high-mountain versus plain areas: Effects on the land surface hydrologic budget in the Alpine area and northern Italy

Claudio Cassardo[1,2,4], Seon Ki Park[2,3,4], Marco Galli[1,a], and Sungmin O[5,b]

[1]Department of Physics and NatRisk Center, University of Torino "Alma Universitas Taurinorum", Torino, Italy
[2]Department of Climate and Energy Systems Engineering, Ewha Womans University, Seoul, Republic of Korea
[3]Department of Environmental Science and Engineering, Ewha Womans University, Seoul, Republic of Korea
[4]Center for Climate/Environment Change Prediction Research and Severe Storm Research Center, Ewha Womans University, Seoul, Republic of Korea
[5]Institute for Geophysics, Astrophysics, and Meteorology, University of Graz, Austria
[a]now at: Air Force Mountain Centre, Sestola, Modena Province, Italy
[b]now at: Max Planck Institute for Biogeochemistry, Jena, Germany
*Correspondence to:* S. K. Park (spark@ewha.ac.kr)

**Abstract.** Climate change may intensify during the second half of the current century. Changes in temperature and precipitation can exert a significant impact on the regional hydrologic cycle. Because the land surface serves as the hub of interactions among the variables constituting the energy and water cycles, evaluating the land surface processes is essential to detail the future climate. In this study, we employ a trusted Soil-Vegetation-Atmosphere Transfer scheme, called the University of Torino model of land Processes Interaction with Atmosphere (UTOPIA), in offline simulations to quantify the changes in hydrologic components in the Alpine area and northern Italy, between the period of 1961–1990 and 2071–2100. The regional climate projections are obtained by the Regional Climate Model version 3 (RegCM3) via two emission scenarios — A2 and B2 from the Intergovernmental Panel on Climate Change Special Report on Emissions Scenarios. The hydroclimate projections, especially from A2, indicate that evapotranspiration generally increases, especially over the plain areas, and consequently the surface soil moisture decreases during summer, falling below the wilting point threshold for an extra month. In the high-mountain areas, due to the earlier snow melting, the land surface becomes snowless for an additional month. The annual mean number of dry (wet) days increases remarkably (slightly); thus increasing the risk of severe droughts, and slightly increasing the risk of floods coincidently. Our results have serious implications on human life, including agricultural production, water sustainability and general infrastructures, over the Alpine and adjacent plain areas, and can be used to plan the managements of water resources, floods, irrigation, forestry, hydropower, and many other relevant activities.

## 1 Introduction

Recent reports from the Intergovernment Panel on Climate Change (IPCC), based on the coupled atmosphere-ocean general circulation models (GCMs) in the condition of increasing concentration of greenhouse gases (IPCC, 2007, 2013), indicate that climate change over the end of this century (e.g., increase of the mean temperature and change of the precipitation amount)

is expected to occur irregularly in space and time but to mostly affect some specific and critical regions (Beniston, 2006), including the vicinity of Mediterranean — well known as one of the world climatic hotspots (Giorgi, 2006; Diffenbaugh and Giorgi, 2012; Gobiet et al., 2014; Vautard et al., 2014; Coppola et al., 2016; Paeth et al., 2017). Within this region, the Alpine and adjacent areas are expected to undergo a relatively larger temperature increase (Giorgi and Lionello, 2008), which has been generally confirmed since the IPCC Fouth Assessment Report (AR4; IPCC, 2007).

In a generic mesoscale basin, such potential changes will influence hydrologic budget; thus, altering the amount of available water and acting as climate feedback. Previous studies conducted over the alpine areas (Giorgi et al., 1997; Beniston et al., 2007; Kotlarski et al., 2015) demonstrated amplification of climate change signal by topography through local hydroclimatic and land surface feedbacks: the snow cycle plays a key role as variations in the cycle of snow pack accumulation and melting affect the generation of snowmelt-driven runoff. In addition, the temperature and seasonal precipitation pattern changes can affect the permanent or seasonal snowmelt, thus affecting streamflow timings, groundwater recharge and runoff, and then again the water availability. Even where the precipitation will increase, the concurrent warming will favor a further increase of evapotranspiration. The decrease of water supplies, conjunctly with the likely increase of the demand, could significantly influence agriculture (the largest consumer of water) and municipal, industrial and other uses (EEA, 2005). Nevertheless, to evaluate locally the net effect of changing climate on water resources, the hydrologic budget must be detailed (Bocchiola et al., 2013).

Seasonal variations of temperature and precipitation also drive changes in runoff and streamflow: for instance, the spring peak streamflow may occur earlier than now in places where snowpack significantly determines the water availability (IPCC, 2007). Such changes may seriously influence the water and flood management, often with significant economic consequences, though the resulting effects may differ for regions even at similar latitudes, as evidenced by Adam et al. (2009) for the high latitudes of North America and Eurasia.

Usually GCMs are calculated in relatively coarse grid spacings; thus inadequately representing the regional topography and climate (Bhaskaran et al., 2012). Therefore, downscaling of the GCM variables to regional scale is essential for better depiction of regional climate: the dynamic downscaling uses the regional climate models (RCMs) with a higher resolution (typically 10–50 km) and the same principles of dynamical and physical processes as GCMs (e.g., Wilby and Wigley, 1997; Christensen et al., 2007; Jury et al., 2015). It is demonstrated that RCMs significantly improve the model precipitation formulation (e.g., Frei et al., 2006; Gao et al., 2006; Buonomo et al., 2007; Boberg et al., 2009). In this context, a project called the Prediction of Regional scenarios and Uncertainties for Defining EuropeaN Climate change risks and Effects (PRUDENCE; http://prudence.dmi.dk/) had been undertaken aiming at providing high resolution climate change scenarios for Europe at the end of the 21st century via dynamical downscaling of global climate simulations (Christensen et al., 2007). Déqué et al. (2005) found that, over Europe, GCMs and RCMs behave similarly for the seasonal mean temperature with higher spread in GCMs; however, during summer, the spread of the RCMs — particularly in terms of precipitation — is larger than that of the GCMs, which indicates that the European summer climate is strongly controlled by parameterized physics and/or high-resolution processes. They also concluded that the PRUDENCE results were confident because the models had a similar response to the given radiative forcing. Déqué et al. (2007) showed that the signal from the PRUDENCE ensemble is significant in terms of the minimum

expected 2 m temperature and precipitation responses. Jacob et al. (2007a) demonstrated that RCMs in PRUDENCE generally reproduce the large-scale circulation of the driving GCM. Coppola and Giorgi (2010) found a broad agreement, in the 21st century climate projections over Italy, between the results obtained from the ensembles of PRUDENCE and the Coupled Model Intercomparison Project (CMIP; https://cmip.llnl.gov/) Phase 3 (CMIP3); however, the CMIP3 GCMs showed a much larger range of bias for temperature and precipitation than the PRUDENCE RCMs. These studies indicate that results from the PRUDENCE and CMIP3/CMIP Phase 5 (CMIP5) experiments are roughly equivalent for the Mediterranean region and the Alpine sector.

The GCMs represent the large-scale atmospheric and oceanic processes. Even if they include sophisticated atmospheric physics and feedbacks with land surface and ocean conditions, they only show conditions averaged over large areas. Hydrologic processes normally operate at quite smaller scales, i.e., meso- and storm scale in meteorology and basin scale in hydrology, and local conditions can be most extreme than those suggested by the areal mean values (see, e.g., the analysis on the groundwater use and recharge in Crosbie et al., 2005). Several recent studies attempted to evaluate the hydrologic effects of climate changes in individual small-scale catchments using a variety of water balance models and climate change scenarios (e.g., Nemec and Schaake, 1982; Gleick, 1986, 1987; Flaschka et al., 1987; Bultot et al., 1988; Ayers et al., 1990; Lettenmaier and Gan, 1990; Klausmeyer, 2005; Buytaert et al., 2009; Berg et al., 2013). Despite of some differences in results, due to the different forcing data or scenarios used (Rind et al., 1992), they have gathered some suitable information at basin or regional scale.

These studies also reveal that the land surface has been recognized as a critical component for the climate. Key points are the partitioning of solar radiation into sensible and latent heat fluxes, and that of precipitation into evaporation, soil storage, groundwater recharge, and runoff. Despite the increased consideration of such processes, the land surface parameters are not systematically measured at neither large scale nor mesoscale, making it hard to perform hydrologic analyses. To overcome such a problem, we have used a methodology called the CLImatology of Parameters at the Surface (CLIPS), proposed by some other studies (e.g., Cassardo et al., 1997, 2009). According to CLIPS, the output of a land surface model (LSM) is used as a surrogate of surface observations, to estimate the surface layer parameters.

The goal of this study is to investigate the effects of climate change, based on high and low emission scenarios, on the hydrologic components in the Alpine and adjacent areas, including the Po Valley in Italy, near the end of this century. Section 2 describes the details of RCM and LSM employed in this study, and Section 3 describes the experiment design. Results concerning the hydrologic budget are reported in Section 4, and conclusions are provided in Section 5.

## 2   Description on models

In this study, calculation of the future hydrologic budget components has been performed through the University of Torino model of land Processes Interaction with Atmosphere (UTOPIA; Cassardo, 2015): meteorological inputs to drive UTOPIA under the current and future climate conditions are obtained from the Regional Climate Model version 3 (RegCM3). Since the details of RegCM3 run have already been published (Giorgi et al., 2004a, b; Gao et al., 2006), here a short description of RegCM3 will be given. Regarding UTOPIA, just some portions relevant for this study are described.

Despite the availability of the products for Europe within the World Climate Research Program COordinated Regional Downscaling EXperiment (EURO-CORDEX; http://www.euro-cordex.net/), which includes a newer version of RegCM (i.e., RegCM4), we decided to employ RegCM3 for the following reasons: 1) RegCM3 had been employed in several important projects, including PRUDENCE, ENSEMBLES (http://ensembles-eu.metoffice.com/), and the Central and Eastern europe Climate change Impact and vulnerabiLIty Assessment (CECILIA; http://www.cecilia-eu.org/), whose outputs had been used in numerous studies focusing on Europe (e.g., Blenkinsop and Fowler, 2007; Christensen and Christensen, 2007; Ballester et al., 2010; Coppola and Giorgi, 2010; Herrera et al., 2010; Rauscher et al., 2010; Kyselý et al., 2011; Torma et al., 2011; Heinrich et al., 2014; Skalák et al., 2014; Faggian, 2015); 2) RegCM3 had also been widely used, even most recently, for the studies of climate projections, model evaluations and sensitivities over the target areas in our study — the Alpine and adjacent areas (e.g., Gao et al., 2006; Smiatek et al., 2009; Coppola and Giorgi, 2010; Im et al., 2010; Coppola et al., 2014; Nadeem and Formayer, 2016; Alo and Anagnostou, 2017); 3) Since a plenty of model outputs were available from several relevant projects (e.g., PRUDENCE, ENSEMBLES, CECILIA, etc.) and we had limited resources for exploring all available data sources, we decided to select a well-known model which had been extensively used for such kind of studies.

## 2.1 RegCM3

The earliest version of RegCM was originally proposed by Dickinson et al. (1989) and Giorgi (1990) to use limited area models as a tool for regional climate studies, with the aim to downscale the GCM results. In this way, the GCMs runs could provide initial and time dependent boundary conditions to RCMs.

The dynamical core of RegCM3 is based on the hydrostatic version of the National Center for Atmospheric Research/Pennsylvania State University Mesoscale Model version 5 (MM5: Grell et al., 1994). The RegCM is a hydrostatic and compressible primitive equation model with a $\sigma$ vertical coordinate. More details on RegCM3 are referred to the MM5 documentation (Grell et al., 1994) and some papers describing previous versions of RegCM (e.g., Giorgi et al., 1993; Giorgi and Shields, 1999). The RegCM3 is documented in Elguindi et al. (2007).

The RegCM3 includes several physical process packages. Precipitation involves both grid- and subgrid-scale processes (e.g., Pal et al., 2000), which are crucial as a source of errors in climate simulations (e.g., Nakicenovic and Swart, 2001). The implemented subgrid precipitation schemes are described in Anthes (1977), Emanuel (1991), Giorgi (1991), Grell (1993), and Emanuel and Živković-Rothman (1999). The physics of the surface processes is described according to the Biosphere-Atmosphere Transfer Scheme (BATS) manual (Dickinson et al., 1993). Subgrid difference of topography and land use are taken into account using a mosaic-type approach (Giorgi et al., 2003b). Two kinds of water bodies are considered – open (e.g., oceans) and closed (e.g., lakes).

The open water bodies are described by the water temperature, introduced as a boundary condition for the model. The closed ones are treated as the open bodies, or using a specific one-dimensional lake model interacting in two-ways with the atmosphere (Hostetler and Bartlein, 1990). Aerosols and chemical compounds are considered accounting to their diffusion and removal processes, and the radiative effects; details about the RegCM3 chemistry are found in Qian et al. (2001), Giorgi et al. (2003a), and Solomon et al. (2006).

The RegCM3 has been employed and tested in various contexts, on various space scales, for a broad range of scientific problems, including climate change (Giorgi et al., 2004a, b; Diffenbaugh et al., 2005; Gao et al., 2006), air quality (Solomon et al., 2006), water resources (Pal and Eltahir, 2002), extreme events (Pal et al., 2004), agriculture (White et al., 2006), land cover change (Abiodun et al., 2007), and biosphere-atmosphere interactions (Pal and Eltahir, 2003).

## 2.2 UTOPIA

The UTOPIA is a diagnostic one-dimensional model, formerly named the Land Surface Process Model (LSPM; Cassardo et al., 1995; Cassardo, 2006). It can be used as a stand-alone basis or be coupled with an atmospheric circulation model or an RCM, serving as the lower boundary condition. All specific details about its use and features are fully described in Cassardo (2015).

The land surface processes in UTOPIA are described in terms of physical fluxes and hydrologic states of the land. The former includes radiation fluxes, momentum fluxes, sensible and latent energy fluxes and heat transfer in multi-layer soil, while the latter includes snow accumulation and melt, rainfall, interception, infiltration, runoff, and soil hydrology. All the fluxes are computed using an electric analogue formulation, in which the fluxes are directly proportional to the gradients of the related scalars and inversely proportional to the adequate resistance.

The UTOPIA domain is vertically subdivided into three main zones – the soil, the vegetation and the atmospheric layer within and above the vegetation canopy layer. Variables are mainly diagnosed in the soil and in the vegetation layers. The canopy itself is represented as a single uniform layer (i.e., big leaf approximation), whose properties are described by vegetation cover and height, leaf area index, albedo, minimum stomatal resistance, leaf dimension, emissivity and root depth. The soil state is described by its temperature and moisture content. These variables are calculated by the integration of heat Fourier equation and conservation of water mass equation using a multi-layer scheme. The main parameters include thermal and hydraulic conductivities, soil porosity, permanent wilting point, dry heat capacity, surface albedo, and emissivity. The UTOPIA can have as many soil layers as a user specifies; however, a sufficient number of layers is required for numerical stability. Note that numerical stability is strictly related to the integration time step — model blows up eventually with an inadequately large time step. This is particularly true in the presence of strong moisture gradients, which could lead to errors in the representation of soil moisture profiles.

Finally, the presence of snow is parametrized with a single layer assumption. Snow can cover separately vegetation and bare soil, and possesses its proper energy and hydrologic budgets; thus interacting with the other components.

The UTOPIA is a diagnostic model; thus, some observations in the atmospheric layer are required as boundary conditions, including air temperature, humidity, pressure, wind speed, cloud cover, long-wave and short-wave incoming radiation, and precipitation rate. Usually these observations are measured values, eventually with the reconstruction of some missing data using adequate interpolation techniques.

The UTOPIA, as well as its predecessor LSPM since 2008, has been tested with field campaigns and measured data either by itself or as coupled with an atmospheric circulation model. Examples of its use can be found in several literatures. Ruti et al. (1997) compared LSPM and BATS in the Po Valley, Italy. Cassardo et al. (1998) studied its dependence on initialization. Cassardo et al. (2005) used LSPM to study surface energy and hydrologic budget on the synoptic scale. Cassardo et al. (2002,

2006) used the LSPM to analyze extreme flood events in Piedmont, Italy. In Cassardo et al. (2007), LSPM has been used to study the 2003 heat wave in Piedmont. Studies with LSPM on non-European climates have also been accomplished, related to very dry sites (Feng et al., 1997; Loglisci et al., 2001), to the onset of the Asian monsoon (Cassardo et al., 2009), and to the soil temperature response in Korea to a changing climate (Park et al., 2017). The UTOPIA was also coupled with the Weather

Research and Forecast (WRF) model, version 3, and applied to a flash flood caused by a landfalled typhoon, as well as to the exceptionally wet period 2008-9 in the northwestern Italy (Zhang et al., 2009; Zhang et al., 2011). Recent applications include studies on the parameterization of soil freezing (Bonanno et al., 2010), and the cold spells over the Alpine area and the Po Valley (Galli et al., 2010). It has also been applied to studies on vineyards environment, including canopy resistance (Prino et al., 2009), energy and hydrologic budgets (Francone et al., 2010), sensitivity to vegetation parameters (Francone et al., 2012a),

and an analysis on turbulence (Francone et al., 2012b).

## 3 Experimental design

The goal of this study is to evaluate the components of the surface hydrologic budget on a mesoscale area from a climatic point of view, and to compare the effects of the climate change on these values. Two 30-years periods have been considered: the first one (1961–1990) is the baseline period or reference climate (RC), whereas the other is the last thirty years of the 21st century

(2071–2100), named the future climate (FC) here. The period 1961–1990 has been employed in numerous previous studies on climate change projections/impacts, even most recently (e.g., Giorgi and Lionello, 2008; Smiatek et al., 2009; Ciscar et al., 2011; Kyselý et al., 2011; Torma et al., 2011; Heinrich et al., 2014; Perez et al., 2014; Skalák et al., 2014; Belda et al., 2015; Dunford et al., 2015; Faggian, 2015; Casajus et al., 2016; Harrison et al., 2016; Gang et al., 2017; Paeth et al., 2017). It had also been used in various climate projection projects using GCMs and/or RCMs, such as CMIP3/CMIP5, PRUDENCE,

ENSEMBLES, and CECILIA.

The climate projections are obtained through the IPCC Special Report on Emissions Scenarios (SRES) A2 and B2 emission scenarios (Nakicenovic and Swart, 2001). Note that the A2 scenario assumes regional resilience and adaptation, while the B2 one assumes still adaptation but local resilience; thus the concentration of carbon dioxide are projected higher for A2 than for B2. The future climates based on the A2 and B2 scenarios are hereafter referred to as $FC_{A2}$ and $FC_{B2}$, respectively.

In the last decade, numerous studies on climate projections/impacts had been conducted using the SRES scenarios, which were the base scenarios in the CMIP3 experiments. After the emergence of new scenarios — Representative Concentration Pathways (RCP; Moss et al., 2010) — which were employed in the IPCC Fifth Assessment Report (AR5; IPCC, 2013) and the CMIP5 experiments, there have been many studies either to check similarities/differences between the two scenario sets for a given projection period (e.g., Riahi et al., 2011; Ward et al., 2011; Rogelj et al., 2012; Matthews and Solomon, 2013; Baker

and Huang, 2014) or to address the value of using both scenario sets for future climate projections (e.g., Peters et al., 2013; O'Sullivan et al., 2016; Nolan et al., 2017).

It turns out that both SRES and RCP scenarios are generally in good agreements, for pairs of closest counterparts, in projecting climate in the 21st century. For example, Riahi et al. (2011) mentioned that SRES A2 was comparable to RCP 8.5.

Ward et al. (2011) found that the RCP 4.5 and SRES B1/A1T scenarios were broadly consistent with the fossil fuel production forecasts. Rogelj et al. (2012) pointed out that the RCP scenarios spanned a larger range of temperature estimates than the SRES scenarios, and indicated similar temperature projections for pairs between the two scenario sets: RCP 8.5 similar to A1FI, RCP6 to B2, and RCP 4.5 to B1, respectively. Matthews and Solomon (2013) showed that the cumulative $CO_2$ emission and corresponding warming at near-term (2030) are approximately the same across all emission scenarios, whereas those at longer-terms (2100) are similar between close counterparts of the selected SRES and RCP scenarios: A1FI to RCP 8.5, A1B to RCP 6, and B1 to RCP 4.5, respectively. Baker and Huang (2014) reported a common drying trend, over the Mediterranean region, between the CMIP3 simulations based on SRES A1B and the CMIP5 simulations based on RCPs 4.5 and 8.5. It is also indicated by Cabré et al. (2016) that SRES A2 has similarities to RCP 8.5 in terms of radiative forcing, future trajectories, and changes in global mean temperature. In Rogelj et al. (2012), differences in warming rates existed between the two scenario sets due to different transient forcings; however, with a 30-year average for each scenario as in our study, the results and conclusions by using the SRES A2/B2 scenarios would not be significantly different from those by using the closest RCP counterparts.

To obtain a broad range of projections, Peters et al. (2013) projected global warming through all available emission scenarios, showing that RCP 8.5 and SRES A1FI and A2 lead to the highest temperature projections. Most recently, O'Sullivan et al. (2016) and Nolan et al. (2017) assessed impacts of climate change on temperature and rainfall, respectively, by mid-21st century in Ireland using both the SRES and RCP scenarios, and provided a wide range of possible climate projections. O'Sullivan et al. (2016) found that future summers had the largest projected warming under RCP 8.5 while future winters had the greatest warming under A1B and A2. Nolan et al. (2017) created a medium-to-low emission ensemble using the RCP4.5 and B1 scenario simulations and a high emission ensemble using the RCP8.5, A1B and A2 simulations, which enabled to have 25 high and 21 medium-to-low emission ensemble comparisons: they found significant projected decreases in mean annual, spring and summer precipitation amounts — largest for summer, with different reduction range for different scenario ensemble.

Furthermore, the SRES scenarios by themselves have often been adopted in most recent studies, even long after the release of the RCP scenarios, because the old scenarios were in accord with their objectives (e.g., Dunford et al., 2015; Jaczewski et al., 2015; Kiguchi et al., 2015; Kim et al., 2015; Casajus et al., 2016; Harrison et al., 2016; Mamoon et al., 2016; Stevanović et al., 2016; Tukimat and Alias, 2016; Zheng et al., 2016; Hassan et al., 2017; Park et al., 2017; da Silva et al., 2017). We employed the SRES marker scenarios because of their long-term consistency in assessing the impact of climate change on global/regional factors of socio-economy and environment during the last decade — including air quality (Jacob and Winner, 2009; Carvalho et al., 2010), water quality/resources (Wilby et al., 2006; Shen et al., 2008, 2014; Luo et al., 2013), energy (Hoogwijk et al., 2005; van Vliet et al., 2012), agriculture/forestry (Lavalle et al., 2009; Calzadilla et al., 2013; Stevanović et al., 2016; Zubizarreta-Gerendiain et al., 2016), fisheries (Barange et al., 2014; Lam et al., 2016), health/disease (Patz et al., 2005; Giorgi and Diffenbaugh, 2008; Ogden et al., 2014), climate/weather extremes (Déqué , 2007; Marengo et al., 2009; Jiang et al., 2012; Rummukainen, 2012), wildfire (Liu et al., 2010; Westerling et al., 2011), ecosystem/biodiversity (Araújo et al., 2008; Feehan et al., 2009; Jones et al., 2009; Fronzek et al., 2012; Walz et al., 2014), and so forth. Although an ensemble approach with all possible scenarios would increase the spread of hydrologic budget simulations, due to the limited resources,

we decided to select two representative marker scenarios: A2 as the higher-end and B2 as the lower-end emission scenario, respectively.

Simulations of RegCM3 for the two periods (i.e., 1961–1990 and 2071–2100) are fully referenced in Giorgi et al. (2004a, b) and Gao et al. (2006), and have been chosen for this study because they are still one of the high-resolution datasets currently available. As shown in Coppola et al. (2016), the RCM outputs with high resolution can allow to efficiently reconstruct the hydrologic cycle at a large-basin scale, even in an orographically complex area such as the Alps.

The domain for this study involves most of the Alpine region and the Po river basin, as shown in Figure 1. It is bordered by the meridians 5°E and 15°E and the parallels 43°N and 48°N. We have chosen this domain for two main reasons: 1) the Alps represent a critical environment that already answered most effectively to the recent climate warming (e.g., Beniston, 2006); and 2) the Alps are the source of the longest and greatest European rivers (e.g., Rhyne, Rhone, Danube, Inn, Arc, Po, etc.). Under these considerations, it is essential to evaluate potential changes in the soil variables and the hydrologic budgets, induced by the climate change.

The RegCM3 outputs are provided on a Lambert grid, with a 20 km spatial resolution, containing 720 land grid points on the analyzed domain (Figure 1). The domain is divided into three sets of grid points in terms of elevation: 1) one representing the plain or low-hill areas lower than or equal to 500 m above sea level (a.s.l.), occupying 34% (blue); 2) another depicting normal mountains between 500 and 2000 m a.s.l., occupying 57% (grey); and 3) the other belonging to the high-mountain areas higher than 2000 m a.s.l., occupying 9% (red). In this study, among all the possible outputs available from UTOPIA, we give particular attention to the state of soil moisture and the components of hydrologic budget — precipitation, evapotranspiration, drainage, and runoff. Note that some of those values were already included in the RegCM3 output database. However, the land surface model of RegCM3 employs an old force-restore method included in the BATS scheme which was demonstrated to be insufficient to properly evaluate hydrologic budget (Ruti et al., 1997). Therefore, we made an offline run with UTOPIA in order to allow a more realistic evaluation of the soil and budget components, and to have a self-consistent set of variables in equilibrium among themselves.

The UTOPIA has been driven using the following output of RegCM3 over each grid point of the domain — precipitation, short- and long-wave radiation, and temperature, humidity, pressure and wind at surface (i.e., the lowest level of RegCM3). This procedure has been used for all three climates (RC, $FC_{A2}$ and $FC_{B2}$).

The UTOPIA has been configured to represent 10 soil layers, following Meng and Quiring (2008) that suggested the use of multiple soil layers to represent well the vertical heterogeneity in soil properties. The thickness of soil layers starts from 5 cm in the top layer, then doubles for every layer going to higher depths. The last layer must be interpreted as a boundary relaxation zone. The soil characteristics have been taken from the ECOCLIMAP database (Masson et al., 2003). No soil-freezing scheme is used, and initial values of soil moisture and temperature have been set following Cassardo (2015).

In terms of vegetation, short grasses are assumed to cover the whole domain. Actually the domain includes the Alps, the Apennines, off-alpine and hilly areas, and plains; thus there is a wide range of vegetation. Regarding plains and hilly areas, vegetation includes pastures, grasslands and some forested areas: mountain areas are mostly covered by trees, and the highest parts are without vegetation or covered by permanent ice (few grid points). We decided to set the vegetation type equal for all

grid points (i.e., short grasses) for the following reasons: 1) for the "reference climate", to avoid any problem in interpretation of results due to the differences in vegetation; and 2) for the "future climate", to alleviate the uncertainty in vegetation type at the end of 21st century. With regard to meteorological variables, this is not a bad assumption because most observation stations are normally installed over short grasses. Moreover, considering plant height, root depth and vegetation characteristics, short grasses can be roughly regarded as most common cereals (wheat, maize, etc.), and would not be quite different from such kind of agricultural products. Finally, we have also performed simulations using the "true" vegetation (as deduced by detailed databases), and the results with the pastures and agricultural areas have generally been confirmed, though the numerical values of the variables were slightly different (not shown).

Although UTOPIA could be driven by the real observations in RC, it is driven by the RegCM3 output in order to keep consistency among the RC and FC simulations and to exclude any possible source of errors caused by differences in input data, irregularity of grid, and/or interpolation of missing observations. In this way, we can compare the FC representations with an analogous RC representation. Thus, here the RegCM3 outputs for each grid point have been used as if they were observed data.

All RegCM3 outputs were available with a time resolution of three hours, and used as input data to UTOPIA. In order to ensure numerical stability of the UTOPIA simulations, these input data, except precipitation, have been interpolated at a rate of one datum per hour: we applied a cubic spline (Burden and Faires, 2004) to the non-intermittent variables like temperature, humidity, and radiation (flux). The intermittent variable like precipitation was simply redistributed to keep its sum, assuming a constant rate: the input data of precipitation was the precipitation cumulated over the timesteps of the RCM output, and could not be interpolated with splines. Although we could have converted precipitation to precipitation rates, interpolated them using splines, and then reconverted to cumulated precipitations over the smaller timestep of UTOPIA, the result of such a complicated procedure was almost equivalent to using the method employed here. Regarding radiation and wind components, we used the splines for the sake of uniformity with other variables. Then, we further controlled some unrealistic values (e.g., negative radiations): we controlled the daily means (or cumulated values) from input (from RegCM3) and output (for UTOPIA) of the interpolation to be non-negative values.

In this study, we employed a single-model approach that has relatively larger uncertainty: it is desirable to employ an ensemble approach, using multiple models and/or initial conditions, to estimate the range of climate projections. Our decision to employ the single-model approach is mainly due to limitation in resources to perform multi-model ensemble simulations for both RCM and land surface model. Given such limitations, a high-resolution single model is often an alternative choice, especially over a complex terrain. Coppola and Giorgi (2010) made a fine-scale (20 km) single-model experiment using RegCM3 and found that both the temperature and precipitation changes via RegCM3 were in line with the CMIP3 and PRUDENCE ensemble results. Generally speaking, multi-model ensembles tend to decrease the errors compared to an individual model; however, due to the averaging operation (e.g., ensemble mean), the spatial and temporal variability of the signal tends to decrease. Moreover, many previous studies on various climate change impacts/projections had been performed using the single-model approach (e.g., Dankers and Feyen, 2008; Beniston, 2009; Im et al., 2010; Krüger et al., 2012; Zanis et al., 2012; Tainio et al., 2013; Park et al., 2017).

The multiple simulations performed for RC and FCs are presented in terms of the temporal and spatial variability by displaying time series (annual cycles) and 2-dimensional maps, respectively, of the mean values of some variables. For time averaging, Xu and Singh (1998) suggested to use monthly mean values for discussing the hydrologic budget variations induced by climate change; however, we preferred a period of 10 days to better quantify time shifts of the physical variables. In this study, the annual cycles are figured via the 10-day averages over the 30-year simulation period, at each elevation-categorized grid-point set. Each month has three 10-day periods: days 1 to 10, 11 to 20, and 21 to the end of the month.

The analyzed variables include precipitation (PR), evapotranspiration (ET), surface runoff (SR), and soil moisture (SM). We noticed that the general trends of annual cycles are similar between RC and FCs. Therefore, in order to accentuate the extent and direction of changes, the future variations of the hydrologic budget components are shown as the differences between FCs and RC; the PR difference ($\Delta$PR) represents $\text{PR}_{FC}$ minus $\text{PR}_{RC}$, where FC is either $\text{FC}_{B2}$ or $\text{FC}_{A2}$ — similarly to $\Delta$ET and $\Delta$SR.

In this study, SM is defined as the quantity of water contained in soil that is composed of solid particles, air and water, and is represented as saturation ratio ($S$):

$$S = \frac{V_w}{V_w + V_a} = \frac{V_w}{V_v},\tag{1}$$

where $V_w$, $V_a$ and $V_v$ are the volumes of water, air and voids, respectively, in soil.

## 4   Results and discussion

In this section, we provide analyses on temporal variability and spatial distribution of hydrologic budget components, making comparisons between RC and FCs. The potential change in dryness/wetness is also assessed through the projection of the number of dry/wet days. Finally, we compare our findings with relevant previous studies, and discuss consistency and uniqueness of our study.

### 4.1   Temporal Variability of Evapotranspiration, Precipitation, Runoff and Soil Moisture

Figure 2 compares the annual cycle of PR, ET and SR in the plain area ($h \leq 500$ m a.s.l.). In the RC summer, ET exceeds PR from the end of June (when ET peaks to about 22 mm) to the end of August (when SM is minimal around 0.52 m$^3$ m$^{-3}$; see Figure 3). PR shows its minimum between mid-June and August, when it is lower than ET. In the RC winter, PR is much higher than ET, and SR exceeds ET from October to March. In the summers of FCs, ET exceeds PR for a longer period (in $\text{FC}_{A2}$), and both scenarios show larger water deficits in July and August, with the PR minimum shifted to August in $\text{FC}_{A2}$ (not shown). Furthermore, the ET maxima shift towards July/August, in both $\text{FC}_{A2}$ and $\text{FC}_{B2}$ (not shown), and the values increase by as much as $3-5$ mm (i.e., $\Delta$ETs).

It is conspicuous that the summer PR decreases in future — between the end of May and the beginning of September in $\text{FC}_{B2}$ (Figure 2b), and between July and September in $\text{FC}_{A2}$ (Figure 2c). On the contrary, PR generally increases in winter, between December and February, in both FCs. In autumn, $\Delta$PRs show large variations in short periods: for instance, in $\text{FC}_{B2}$,

it varies as $-6$ mm in mid-September, $+10$ mm in late September, $-12$ mm in late October, $+15$ mm in mid-November, and $-7$ mm in late November. Regarding $\Delta$ET, there are almost no variations in cold months, while there is a small increment (up to 3 mm) between April and September in $\mathrm{FC}_{B2}$, and a larger increment in the same period in $\mathrm{FC}_{A2}$, with the largest value in August ($\sim$5 mm). This large variation in PR is partly due to orographic effect. As reported by Gao et al. (2006), in winter the southwesterly flow increases across the Alps, and causes a maximum of precipitation increase over the southern Alps; in autumn the main circulation change is in the easterly and southeasterly direction.

Figure 3 shows the 10-day mean values of SM for the plain area, expressed as saturation ratio — see Eq. (1). Variations of SM in plains are almost negligible in a colder period (late November – mid-May), but are large during a warmer period (late May – mid-November): the driest points are antedated by $\sim$10 days in FCs, still being in August, and their values decrease by $\sim$0.1 m$^3$ m$^{-3}$. The decrease begins already in spring (from late May) and continues till late October ($\mathrm{FC}_{B2}$) or early November ($\mathrm{FC}_{A2}$), with the largest depletion in early August ($\mathrm{FC}_{B2}$) and in early to mid- August ($\mathrm{FC}_{A2}$). Moreover, the period that future SM values are lower than the lowest SM of RC (i.e., $\sim$0.52 m$^3$ m$^{-3}$ in mid-August) extends from early July to early September in $\mathrm{FC}_{B2}$ and to mid-September in $\mathrm{FC}_{A2}$. In the driest periods of FCs, several grid points in the plains go below their permanent wilting points (PWP), which vary according to soil type, or remain below PWP for an excessive duration by about one month. Our results regarding the future changes of SM in the warm period — an increase in days of SM lower than the lowest SM of RC, and a surplus of period below PWP — signify that, if the land use of the grid points is pasture, we need appropriate countermeasures to ensure an adequate productivity. During the cold period in plains, SM shows the highest values ($\sim$0.73 m$^3$ m$^{-3}$) in both RC and FCs; the SM values of FCs slightly exceed those in RC, due to the small increments of PR in this period (see Figures 2b and c).

Figure 4 shows the annual cycle of hydrologic budget components over the high-mountain area ($h > 2000$ m a.s.l.). In both RC and FCs, PR does not exceed ET while the gap between the two variables narrows in the FC summers, due to an increase in ET and a decrease in PR. In RC, ET peaks in mid-July while PR peaks in late June. The peak of SR, between May and June, is out of phase because it is also affected by the concurrent snow melting. It is noteworthy that PRs in summer and fall generally decrease in FCs (i.e., $\Delta$PR $< 0$) from mid-June to mid-November: except for short terms in early July, from mid- to late August and from late September to late October in $\mathrm{FC}_{B2}$, and except only from early October to early November in $\mathrm{FC}_{A2}$. On the contrary, in winter and spring, PRs generally increase in FCs from mid-January to early June except for short-term decreases in mid-April and mid-May. Regarding $\Delta$ET, there are almost no variations in cold months, as expected (due to snow cover), whereas there is a large increment ($\sim$10 mm) between May and June, and a low-to-moderate increment ($\sim$2–6 mm) between July and October in FCs.

Finally, for $\Delta$SR at high mountains, there is a weak increase ($< 5$ mm) between late November and late March, a stronger increment ($\sim$10 mm) in April, especially in $\mathrm{FC}_{B2}$, a strong decrease (up to $-25$ to $-31$ mm) between May and June, and a general weak decrease in summer between July and September (see Figures 4b and c). As a result, the maxima of SR in FCs significantly decrease and their occurrence dates shift ahead to May for $\mathrm{FC}_{B2}$ and between April and May for $\mathrm{FC}_{A2}$ because snow melting occurs nearly 30–40 days earlier (see Cassardo et al., 2018) — see also the analysis on frost frequency in Galli et al. (2010). Coppola et al. (2016) also reported that, regarding the 75th percentile in the Alpine areas, the snowmelt-driven

runoff timing moves earlier by about 35 days — due to the largest decrease of snow cover between April and June, sustaining the spring runoff maximum. Those variations in our result are in line with the changes of snowpack in FCs, which starts to melt earlier, between late April and early May. We should consider that changes in snow cover affect the surface energy budget through the snow-albedo feedback mechanism (Giorgi et al., 1997); that is, a reduction of snow cover decreases the surface albedo, and thus increases the absorption of solar radiation at the surface, resulting in warming. Moreover, soil temperature starts rising earlier in the year at the snowless areas. Our results also agree with other studies, carried out using RCMs over the Alpine areas: for example, Lautenschlager et al. (2008) for PR and ET, and Jacob et al. (2007b) for snow. Note that SR in RC is almost null between mid-December to March while $\Delta$SRs in FCs in the same period are positive: this indicates the presence of rainfalls and/or snow melting over at least some parts of the high-mountain grid points, even in the coldest periods.

Figure 5 shows SM at the high-mountain grid points and demonstrates the effects of hydrologic budget components on surface SM. We note that the behaviors of SM at high mountains are substantially different from those at plains (cf. Fig. 3). In RC, the highest SM ($\sim$0.65 m$^3$ m$^{-3}$) occurs at early June while the lowest SM ($\sim$0.51 m$^3$ m$^{-3}$) arises at early to mid- March. The increase in SM from late March to early June is related to snow melting due to increase in net radiation. Surface SM in RC starts to decrease as the cold season starts in early November, reaching the minimum in mid-March. Note that SMs during the same cold period in FCs are larger than SM in RC, evidencing a larger amount of liquid precipitation in FCs: in other words, winter rainfalls will be more frequent in the future. The peak of SM in spring is advanced by 10–20 days in FC, occurring in early May. The magnitude of maximum SM in FC is a bit lower than that in RC but the spread is larger, implying that snow ablation starts much earlier and lasts longer. In addition, the occurrence of the minimum SR shifts from mid-March in RC to summer in FC: in both February and early August (i.e., two minima) in FC$_{B2}$, and late August in FC$_{A2}$. This shifting is mainly caused by the enhancement of ET.

## 4.2 Spatial Distribution of Evapotranspiration, Precipitation, Runoff and Soil Moisture

Our analyses illustrate that the differences in the SM behaviors between RC and FC, at both plains (Figure 3) and high mountains (Figure 5), are strongly linked to the variations of the hydrologic budget components. In this section, to understand such linkage more clearly, we perform analyses on the spatial distribution of hydrologic variables (i.e., PR, ET, SR and surface SM) along with discussions on the associated energy variables (i.e., net radiation (NR) and surface soil temperature (ST)), during summer when such variables generally show their largest values. Details in the analyses of NR and ST are referred to Cassardo et al. (2018).

Figure 6 shows the variables averaged in the month of July, in which PR and surface SM are close to their annual minima while ET is close to its annual maximum. Here, we discuss the variables in terms of anomalies of FC$_{A2}$ only because of similar patterns to but larger variations than those of FC$_{B2}$. Variables in Figure 6 are anomalies of hydrologic budget components: $\Delta$ET, $\Delta$PR, $\Delta$SR and $\Delta$SM where, e.g., $\Delta$ET represents ET$_{FC_{A2}}$ − ET$_{RC}$.

Compared to RC, we notice a large increment of NR everywhere in FC$_{A2}$ (not shown), with the exception of few grid points located in the central and western Alps. Regarding $\Delta$ET (Figure 6a), plains along the Po River and the northern off-alpine regions (i.e., middle-slope and/or foot) show the largest increments, well correlated to $\Delta$NR, implying that most of the

available energy excess is used for evaporative processes. In contrast, on the Apennines and central Alps, $\Delta$ETs are almost null or slightly negative while $\Delta$NRs are insignificantly positive. $\Delta$PR (Figure 6b) and $\Delta$SR (Figure 6c) show similar signals, with a general deficit, especially on the eastern and western Alpine areas. In particular, consistent with Coppola et al. (2016), $\Delta$PR depicts a dipolar pattern, especially on the eastern part of the Italian Alps, with positive values over the Alps and its north and negative values over south of the Alps. Surface $\Delta$SM (Figure 6d) shows a general reduction, larger in the zones at latitudes lower than 45°N, whereas surface $\Delta$ST (not shown) is almost uniformly larger in the considered domain. As ETs increase (i.e., $\Delta$ET > 0), SMs generally decrease; however, both decrease over some regions where $\Delta$SMs are strongly negative — on the western mountainous Emilia Romagna region and Toscana, and along the Po River and in central and southern Piemonte as well (cf. Figures 6a and 6d). When SM decreases below the wilting point, evaporation generally ceases because there is no available water for further ET, and the ET anomaly (i.e., $\Delta$ET) can be negative. Considering that most of those areas are important for agricultural production (see, e.g., Prino et al., 2009, a study on grapevine in Piemonte region), our results constitute a threatening challenge for future agricultural productivity.

It is evident that $\Delta$ET and $\Delta$PR do not show a linear correlation (cf. Figures 6a and 6b). $\Delta$ETs are generally positive, whereas $\Delta$PRs are distributed around null with some positive peaks on the Apennines and northwestern Italy and large negative peaks on some Alpine locations. This disparity brings about and/or enhances the nonlinear interactions among temperature, evaporation, soil moisture, etc. Noting that nonlinearity can develop even with small perturbations (e.g., Park, 1999), our results elucidate that similar investigations can only be conducted using models that are able to give a correct estimation of energy and hydrologic processes.

### 4.3 Number of Dry and Wet Days in the Future Climate

The availability of the SM estimations enables us to evaluate the occurrence of dry and wet days, instead of using atmospheric relative humidity as usual, in a similar way to figure the warm and cold days via the ST estimations. We employ SM to assess the dry and wet days in FCs because we consider it as a more valuable indicator of the soil hydrologic conditions, directly reflecting the hydrologic status of the soil water, e.g., used by plants. Here, we limit the analysis to the surface SM (i.e., in the top soil layer with a depth of 5 cm), due to its significant impact on several agricultural productions.

Actually, for the short grass vegetation category considered in our simulations, the root layer is only 5 cm deep, as the grass is only 10 cm high. Despite this value seems too low, it represents the typical height for the landscapes of Po Valley (at least in its portion occupied by natural vegetation). Furthermore, the upper soil layer represents the greatest effect of the atmosphere-land surface-soil interactions. Given that we are interested in the present versus future hydrologic budget components, it is appropriate to focus on the top soil layer, where the most dynamic interactions with atmosphere and land surface occur. More specifically, the water content of the soil layer that represents the largest variations of moisture is subjected to direct evaporation, to the transpiration from vegetation roots, to the gravitational drainage to the second soil layer, to the capillary suck of moisture from the second soil layer, and finally to the eventual precipitation, eventual vegetation drainage, and eventual snow runoff.

In order to find the absolute thresholds for SM, we have selected two parameters: PWP and the field capacity. PWP is the SM level below which the osmotic pressure of the plant roots is insufficient to extract water from the soil, and is usually considered

as an indicator of a serious water deficit for agricultural practices. The field capacity represents the SM level above which the gravitational drainage, due to soil hydraulic conductivity, causes a rapid removal of the excessive water through percolation into deeper layers; thus it is considered as a threshold above which soil is very wet, as in the cases of very intense precipitations, sometimes causing floods. Since these two values change according to the soil type and texture, we define a non-dimensional index, $Q_I$, which is independent from soil type, as:

$$Q_I = \frac{q_1 - q_{wi}}{q_{fc} - q_{wi}} \qquad (2)$$

where $q_1$ is the moisture of the top soil layer, $q_{wi}$ is PWP and $q_{fc}$ is the field capacity. All the values are expressed in the unit of soil saturation ratio. In this way, the soil wetness is categorized in terms of $Q_I$ as: extremely dry soil for $Q_I \leq 0$, and extremely wet soil for $Q_I \geq 1$. In this study, we define the thresholds for dry soil and wet soil as $Q_I = 0$ and $Q_I = 0.8$, respectively. Note that it is quite rare to see the cases with $Q_I = 1$ because the 3-hourly precipitation data from RegCM3 are interpolated to hourly data by keeping the constant rain rate, to be used as input for UTOPIA. Therefore, we have arbitrarily defined the threshold for wet soil as $Q_I = 0.8$.

Figure 7 shows the anomalies of dry and wet days in FC$_{A2}$. The number of dry days generally increase in most of the domain except the Alpine high-mountain areas (Figure 7a). Higher number of dry days (e.g., 30–50 days) occur over the regions of extreme soil dryness — the coastal areas as well as the off-alpine regions of the Alps and the Apennines (cf. Figure 6d). The interannual variability of the dry-day occurrence also decreases (not shown), implying that our results are relatively robust and that we may experience drought over the non-high-mountain areas in almost every year.

The number of wet days, on the other hand, is almost stationary over plains but increases by 10–15 days in some localized regions close to the Alps in the Italian side (especially in the Lombardy region), and by even more than 20 days at the feet of the Alps in Switzerland, France and Austria (Figure 7b). The interannual variability is generally stationary, but increases in the areas with the largest numbers of wet days (not shown). Therefore, in FC$_{A2}$, we can have more occasions of reaching high values of surface SM, hence potentially higher risk of floods. This also implicates corresponding higher possibility of hydrogeological instability over the same areas of higher flood risk.

Overall, in the plain areas including the Po Valley, $\Delta$ET is positive while $\Delta$PR is weekly negative and $\Delta$SM is moderately negative (especially during summer as in Figs. 2 and 3). With more significant overall increases in NR over plains, the combined effect will bring about larger evaporation and lower soil moisture, thus overall increase in the number of dry days, mostly attributed to much drier climate in summer. Meanwhile, over the high-mountain areas, PR, SR and SM increase while ET shows little variation in spring and winter (see Figs. 4 and 5). As SM is large over high mountains, we have more source of atmospheric moisture through evaporation there. Then, through the combined effect of terrain-induced convective motion, increase in NR (though less significant) and pre-existing snow, we can have more snow melting (during spring) and more liquid precipitation (especially during winter), resulting in more wet days, again mostly attributed to much wetter climate in winter.

## 4.4 Comparative Discussion on Previous Works

The Mediterranean basin is recognized as one of the climatic hotspots around the world (Giorgi, 2006; Diffenbaugh and Giorgi, 2012; Gobiet et al., 2014; Vautard et al., 2014; Paeth et al., 2017). The Alps and its adjacent areas therein, including the Po River basin in Italy, have been a target region of many climate projection studies, using either a single RCM or an ensemble of GCMs/RCMs (e.g., to mention just a few, Gao et al., 2006; Giorgi and Lionello, 2008; Im et al., 2010; Dobler et al., 2012; Shaltout and Omstedt, 2014; Addor et al., 2014; Coppola et al., 2014; Gobiet et al., 2014; Torma et al., 2015; Coppola et al., 2016; Frei et al., 2018).

In general, those studies showed good agreements with our results and produced consistent results of climate projections at the end of 21st century over the study region. However, none of them studied hydroclimate projections of full water cycle by assessing all hydrologic components — precipitation, evapotranspiration, runoff and soil moisture — as in our study. Most them focused on just some specific component(s) of water cycle, e.g., precipitation and/or surface runoff. For instance, Giorgi and Lionello (2008) studied climate change projections for the Mediterranean region, focusing on precipitation and temperature; Coppola et al. (2014) studied the impact of climate change on the Po basin, addressing discharge; and Torma et al. (2015) carried out ensemble RCM projections over the Alps, centering about precipitation.

Nevertheless, it is meaningful to compare our findings, on overall hydrologic components, with other studies over the same study area. Basically, most of previous studies showed consistent results with ours, as exampled in the followings. Gao et al. (2006) illustrated the positive anomaly of precipitation in future climate over the southern Alps from autumn through spring, and the negative anomaly in summer over the highest peaks of the Alps. Giorgi and Lionello (2008) remarked the peculiar behavior of the Alpine region, compared to the Mediterranean basin, with moderate drying during warmer seasons and increase of precipitation in winter, and a large increment of interannual variability, which can lead to an increase of extreme events such as droughts and floods. Im et al. (2010) discussed a surrogate climate change simulation over the Alpine region and found that the winter precipitation increased with a significant dependence on elevation while the summer precipitation decreased over the Alpine mountain chain, due to a local surface-atmosphere feedback mechanism involving reduced snow cover and soil moisture at the beginning of summer. Dobler et al. (2012) showed that future precipitation decreased during summer and increased during winter and spring over the Alps; Shaltout and Omstedt (2014) also noted the increment of winter precipitation in future climate in the Alpine region, due to a negative correlation with decreasing pressure patterns. Addor et al. (2014) addressed the larger changes of precipitation regime in the higher-elevation Alpine catchments. Coppola et al. (2014) examined the variation of the discharge maxima of the Po river in future climate, and concluded that the winter-spring maximum would increase and the summer-autumn maximum will decrease. Torma et al. (2015) confirmed that future precipitation would increase (decrease) over northern (southern) Europe, with most of the Alpine region exhibiting a positive (negative) precipitation change in the winter (summer). Frei et al. (2018) found a robust signal of decreasing snowfall amounts, from September to May, over most parts of the Alps, with relative changes in mean snowfall being strongly dependent on elevation. In a review paper based on the existing literature and additional analyses on climate change in the Alps, Gobiet et al.

(2014) concluded that warming induces a seasonal precipitation change — increase in winter and decrease in summer — and a drastic decrease of snow cover below 1500–2000 m in the Alps.

Compared to most other studies, which focused on the subcomponent(s) of hydrologic cycle, our study is quite exhaustive and has its own uniqueness: our study provides more complete analyses on all hydrologic components, including soil moisture, for both reference climate and future projections. Furthermore, along with a study on the land surface energy balance (Cassardo et al., 2018), we provide discussions on the linkages between the hydrologic and energy components to complete the full description of hydroclimatic changes. These enable us to better quantify some significant variations in the frame of changing climate in the Alpine and adjacent areas, in which the climatic change shows a larger variability.

## 5 Conclusions

In this study, we investigated the characteristic changes of hydrologic budget components and soil moisture, over the Alpine areas and northern Italy under the projected conditions of future climate (FC; 2071–2100), compared to the reference climate (RC). We employed the University of TOrino model of land Processes Interaction with Atmosphere (UTOPIA) in offline simulations. The meteorological input data in FCs are provided by the Regional Climate Model version 3 (RegCM3), based on the A2 and B2 scenarios from the Intergovernment Panel on Climate Change Special Report on Emissions Scenarios.

In FCs based on the A2 and B2 scenarios ($FC_{A2}$ and $FC_{B2}$, respectively), the most significant changes are the increment of evapotranspiration (ET) and the subsequent depletion of soil moisture (SM), more remarkably in $FC_{A2}$. Precipitation (PR) shows the lowest values while ET depicts the highest values in the future summer (in particular, July), when SMs are the lowest in many grid points. In the plain area, the minimum SM in FC occurs about 20–30 days earlier than in RC, and remains low for the successive months up to November. In the high-mountain area, the surface runoff (SR) coming from the snow melting keeps the soil water amount sufficiently high to maintain the ET levels high from May to October, especially in $FC_{A2}$; thus, ET (or latent heat flux) always exceeding sensible heat flux (SHF). In plains, the period in which ET exceeds PR elongates by about one month, mainly in spring. Moreover, SM decreases also for one more month in summer, falling below the wilting point threshold in the surface soil layer. In high mountains, due to the earlier occurrence of snow melting, the land surface becomes snowless for an additional month.

We found that these changes in the hydrologic budget components are strongly related to the variations of net radiation (NR), which generally increase in the Alpine area, causing the warming of both the top soil layer and the soil surface — the former through an enhanced SHF, and the latter due to the highest soil heat flux (see, e.g., Cassardo et al., 2018). Under the future conditions of increasing NR and soil temperature along with decreasing SM, we expect two climatic feedbacks to take place: 1) a drier soil brings about higher albedo, and 2) a warmer soil emits more long-wave radiations. Both feedbacks act to decrease NR eventually — i.e., negative feedbacks. However, there are coincident increments of SHF to the atmosphere as well as longwave radiation emitted by the warmer atmosphere. The overall outcome cannot be generalized because it depends on the intensity of individual component of the energy and hydrologic budgets. This confirms that the climate system is quite complex

and that, to evaluate well the surface conditions, it is essential to calculate the energy and hydrologic budget components in detail.

The values presented in this study refer only to the average conditions; however, considering the large interannual variability of hydrologic variables registered over those areas in RC, we expect to have more frequent and intenser occurrences of longer dry spells (hence severe droughts) and heat waves in FCs, especially in middle summers. As most agricultural products intensively grow in summer (e.g., wheat, rice, maize and grapevine, and other typical products in the Po valley), the potential conditions of elongated drought will significantly exert unfavorable impacts on agricultural production (Bocchiola et al., 2013). Other activities related to water supply (e.g., industry, hydroelectric power production, etc.) can also suffer serious problems, consequently exerting harmful impacts on economy and human health in local regions.

On the contrary, during winter, PRs generally increase in FCs, with a larger number of the liquid precipitation events at high elevations. Furthermore, in spring, snow melting occurs earlier by about one month, thus resulting in precedence of the SR peak by about 20–30 days. In winter, the SR amount generally increases. By taking into account the large interannual variability of PR, this runoff increases the occurrence and/or duration of wet periods (e.g., heavy rainfalls and floods) during winter and spring in FCs.

We also examined potential changes in the number of dry and wet days in $FC_{A2}$ by analyzing surface SMs. Our results report a higher possibility of having SMs below the wilting point in the plain and coastal areas, and a probability of slightly increasing wet days, particularly in the off-alpine areas.

We note that the numerical values of all variables are dependent on the performance of employed model. Noting that our study is based on a single-model approach, uncertainties in the projected changes related to model bias and ensemble variability can be large; thus our results should be interpreted with caution. In this context, further research is needed to obtain more robust results from an ensemble approach.

Recent studies demonstrate that the accuracy of land surface processes diagnosed by land surface models can be further improved by considering various aspects of vegetation effects in the subgrid-scale parameterizations (e.g., Park and Park, 2016; Gim et al., 2017). Moreover, the model uncertainties can be significantly reduced by optimal estimation of the parameter values in the schemes (e.g., Lee et al., 2006; Yu et al., 2013) and/or seeking for an optimized set among multiple-physics optional schemes (e.g., Hong et al., 2014, 2015). By applying these methods, the details of model-generated spatial/temporal changes in the future energy and hydrologic budgets can be different from the current results; however, we believe that the general trends are not significantly disparate. Overall, our findings can provide a useful guideline to plan the managements of water resources, floods, irrigation, forestry, hydropower, and many other activities relevant for human life.

*Data availability.* The ECOCLIMAP data is available online from https://opensource.umr-cnrm.fr/projects/ecoclimap.

*Competing interests.* The authors declare that they have no conflict of interest.

*Acknowledgements.* The authors acknowledge the Earth System Physics Section of the ICTP, Italy, for providing the RegCM3 dataset. S. O was partly supported by the University of Torino (UNITO) for visiting its Department of Physics under the World Wide Style grant. C. Cassardo and S. K. Park are supported by the governments of Italy and Korea, respectively, for visiting each institution for collaborative research via the bilateral scientific agreements. This work was partly supported by the National Research Foundation grant (No. 2009-0083527) funded by the Korean government (MSIP). The work is partially done during a sabbatical leave by S. K. Park to UNITO in 2017.

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

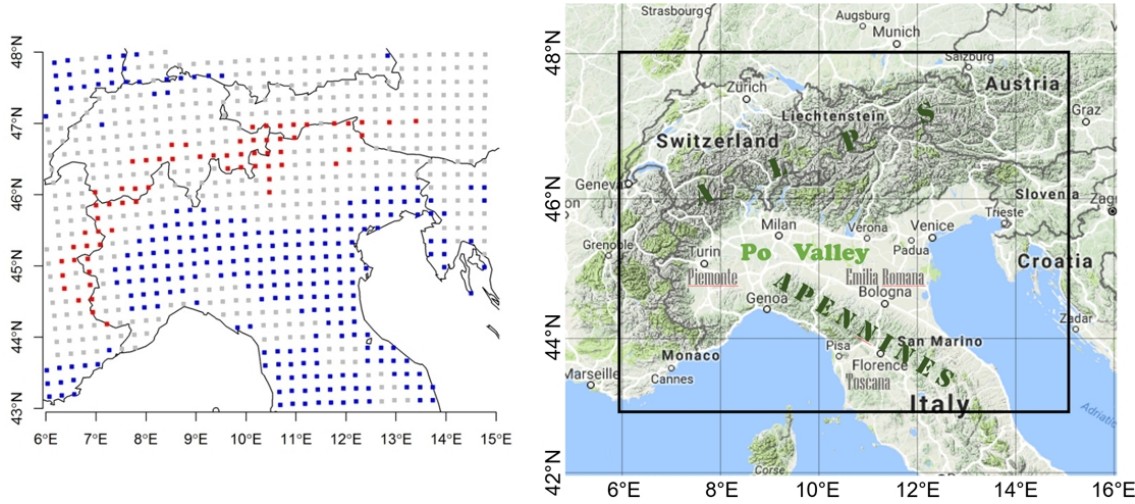

**Figure 1.** Computational domain with grid points (left panel) and geographic map (right panel) with boundary of the study area (black solid lines). Grid points represent, in terms of the grid elevation ($h$), the plain area ($h \leq 500$ m a.s.l.; blue), the normal mountains ($500 < h \leq 2000$ m a.s.l.; grey), and the high-mountain area ($h > 2000$ m a.s.l.; red). The map on the right panel is modified from the Google Maps.

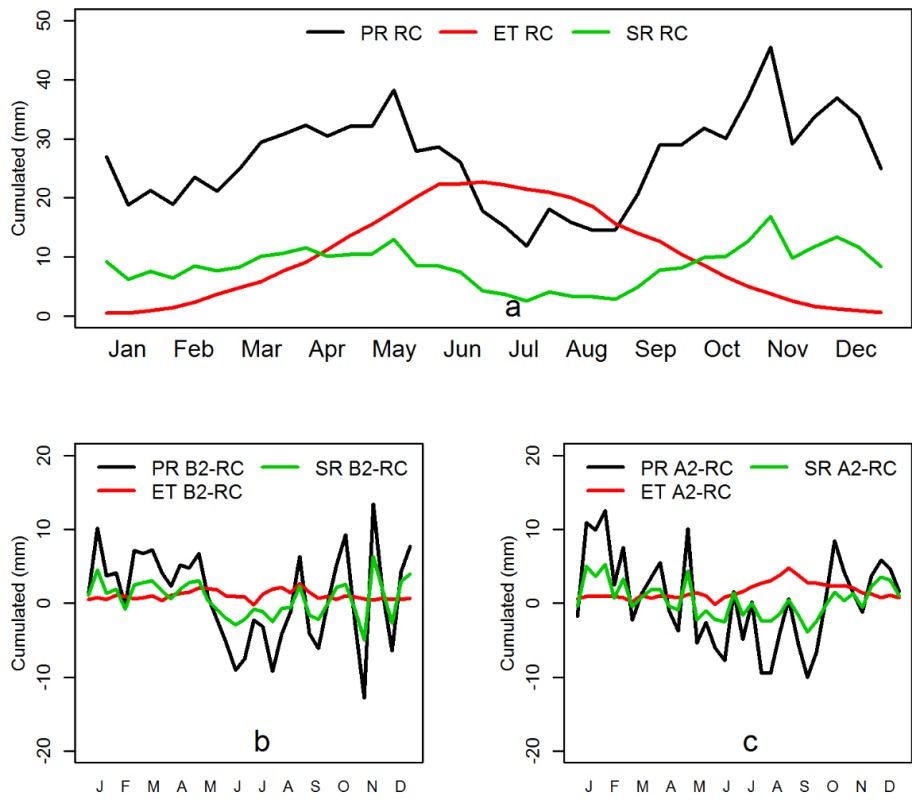

**Figure 2.** Annual cycles of the 10-day average values of the surface hydrologic budget components for the plain area for a) RC, b) $FC_{B2} -$ RC, and c) $FC_{A2} -$ RC. Here, PR is precipitation, ET evapotranspiration, and SR surface runoff. Units are mm.

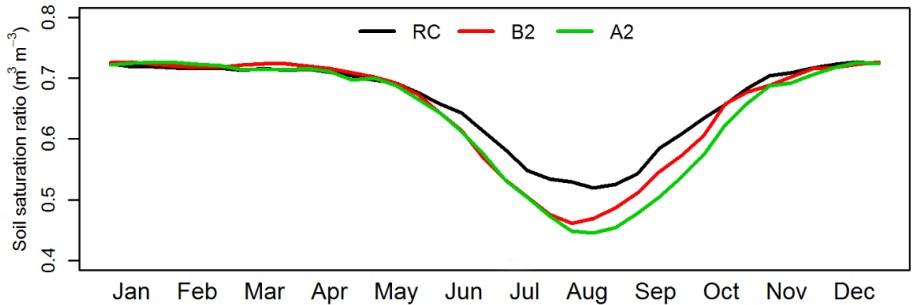

**Figure 3.** Annual cycles of the 10-day average values of SM, expressed as saturation ratio (in $m^3 \ m^{-3}$), at the soil surface layer (a depth of 0.05 m) in RC, $FC_{B2}$, and $FC_{A2}$ for the plain area.

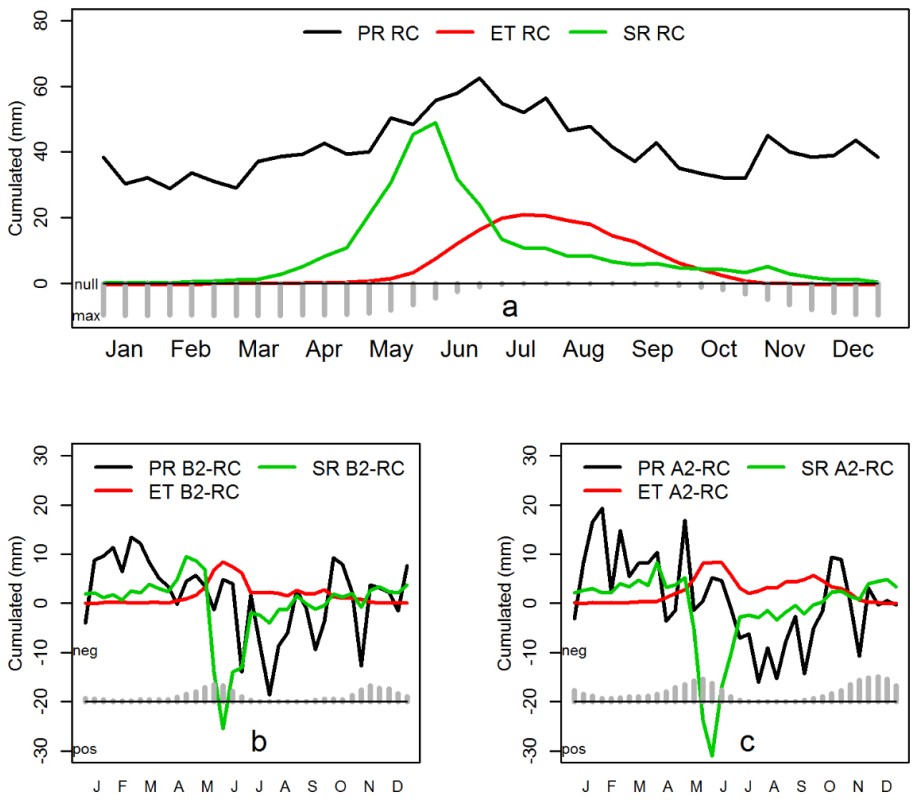

**Figure 4.** Same as in Figure 2 but for the high-mountain area. Grey bars at the lower portion in a) represent the snow cover (in m) in RC varying from 0 m (null) to 1 m (max); in b) and c) the snow cover difference (in m) between the corresponding FC and RC varying from −1 m (neg) to 1 m (pos). The periods of snow ablation (late spring) and accumulation (mid or late autumn) are well identified.

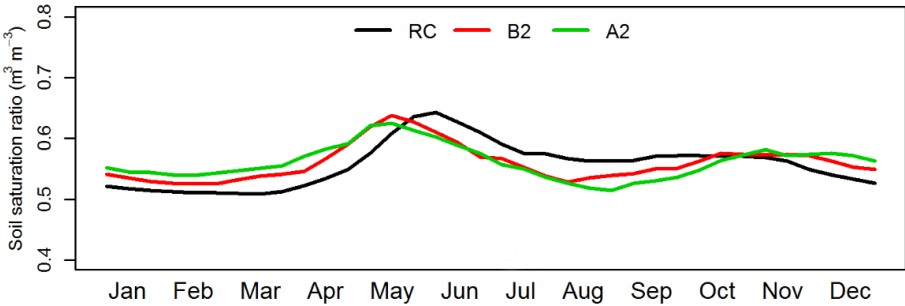

**Figure 5.** Same as in Figure 3 but for the high-mountain area.

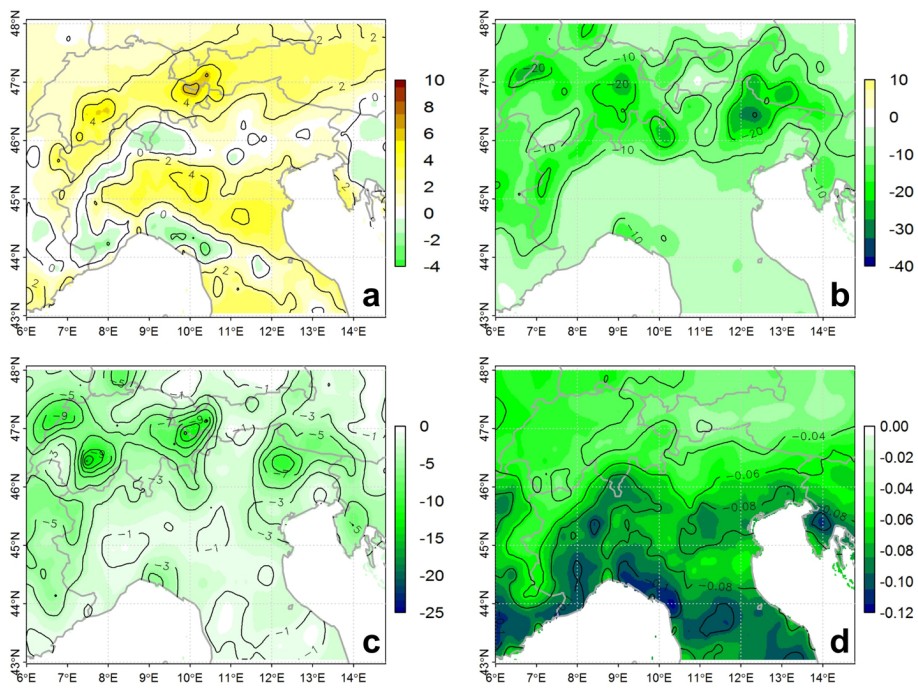

**Figure 6.** Hydrologic budget components: differences between $FC_{A2}$ and RC (i.e., $FC_{A2} - RC$) of the mean values of a) ET (in mm), b) PR (in mm), c) SR (in mm) and d) surface SM (in m$^3$ m$^{-3}$). The mean is calculated over the month of July.

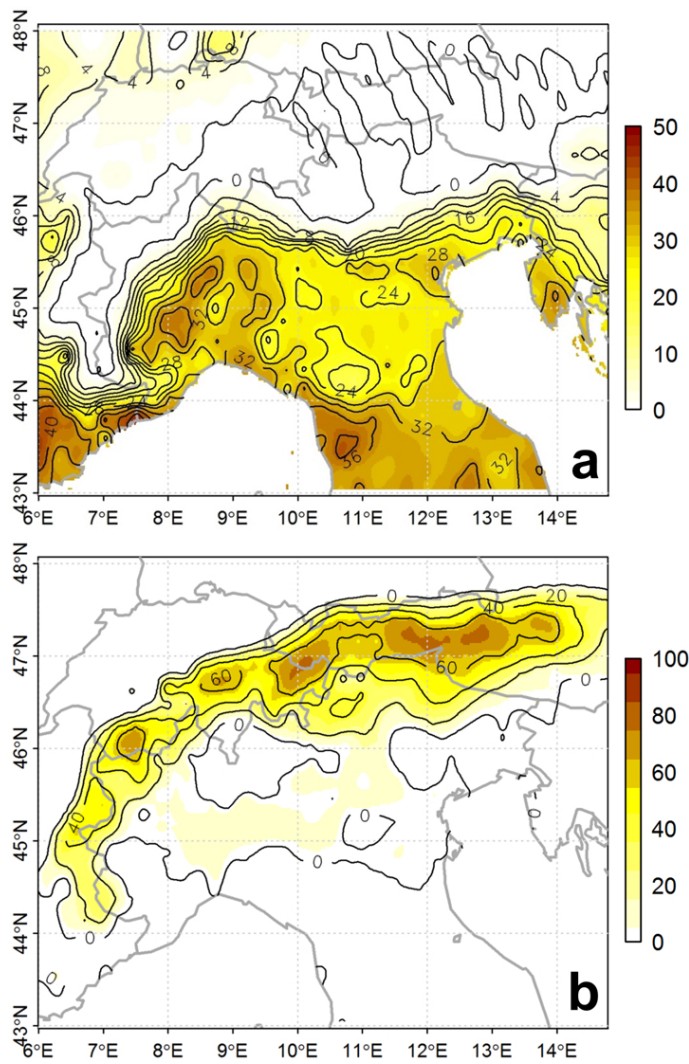

**Figure 7.** The anomalies in number of a) dry days and b) wet days in FC$_{A2}$.