# Peer review of "Climate change over the high-mountain versus plain areas: Effects on the land surface hydrologic budget in the Alpine area and northern Italy"

_Hydrology and Earth System Sciences, 2017_

## Referee Comment (RC1) · Anonymous Referee #1 · 21 Nov 2017

**Major comments**

- The paper lacks any comparison with other papers. What is improvement in using the PRUDENCE output with respect, say, to other global and regional Earth system models? I guess, at least some comparison with the CMIP3/CMIP5 simulations should be included.

- How specific are the UTOPIA simulations with respect to other land surface models? At least, some peculiarities of this model should be discussed in the context of other land surface models.

[Figure]

- Why the CMIP3 scenarios SRES A2/B2 are used? Now they are superceded by the RCP scenarios family.

- And the most important one: Which new knowledge does the paper deliver to us?

**Minor comments**

- ll. 6 and 7: I guess 'FC' in the abstract is difficult to understand. Please be more specific (e.g. replace it by time interval '2071-2100').

- l. 10: 'Annual or seasonal variations..' What is the difference between them? Probably, authors mean 'interannual' in place of 'annual'?

---

## Referee Comment (RC2) · Anonymous Referee #2 · 30 Nov 2017

The manuscript 'Climate change over the high-mountain versus plain areas: Effects on the land surface hydrologic budget in the Alpine and northern Italy' presents climate change impacts on evapotranspiration, precipitation and soil moisture over the Alpine and northern Italy using regional climate model and land surface process model. Authors well deliver the changes of hydrological budget under climate change. However there are a few concerns described below.

Major comments 1. Authors used RegCM3 in this study. Newer version model dose not necessary mean that having better performance but authors need to justify why older version model with older scenario (AR4) was employed in this study. 2. In this study,

[Figure]

authors employed single RCM and single land model. Authors need to discuss about model uncertainties comparing to the multi-model approaches. 3. I do believe there are quite a few previous studies over the study region. Authors need to introduce them. 4. Authors discussed about possible agricultural impact due to the lower soil moisture. However, in this study, vegetation type in UTOPIA were set single type of vegetation (short grasses) all over the domain. Have you done any sensitivity test on vegetation types?

Minor Comments. 1. Page 1. Remove all acronyms in abstract. 2. Page 2, line 15. Global circulation models. Is this different one as GCMs previously defined? 3. Page 3, line22-24. Related with major comments 1. EURO-CORDEX has RegCM4 with higher resolution. 4. Page 4, line 5-19. Move to methods section. 5. Page 4, line 20. Energy variables are critical part of your argument. Please include the figures as a supplementary. 6. Page 4, line 25. I cannot find the PR minimum shifts in the figure. 7. Page 4, line 26. ET of FCB2 shows double peaks rather than shifting in Fig2b. 8. Page 4, line 29-31. It looks like the large variation stems on future PR variation. Can you explain why PR has large variation? 9. Page 6 line 27-29. Can you include the names of geographical location on the map (e.g. Fig. 1)

---

## Referee Comment (RC3) · Anonymous Referee #3 · 3 Dec 2017

The paper presents assessment of changes of land water budget terms in Northern Italy under future climate changes. The regional climate model RegCM3 simulations are used as a forcing for the land surface scheme UTOPIA. The modeled seasonal and spatial patterns of precipitation, evapotranspiration, runoff, soil storage, net radiation are examined, and implications for regional economies are formulated.

My major concerns related to the paper are: 1) There is no proper comparison of results obtained to other similar studies conducted for this region, elucidating what is the new knowledge attained. Some of other relevant papers are cited (Lautenschlager et al., 2008; Jacob et al., 2007), but the comparison is very limited. 2) The physical

analysis of simulation results is somewhat superficial. The simple effects are explained, whereas the more complicated ones (like the absence of spatial correlation between evapotranspiration and precipitation, lines 1-4 on p.7) are commented by too general statements. In this respect, the striking separation of regions with large dry and wet days numbers anomalies at Figure 7 is left without deserving physical analysis (lines 25+, p.7 are merely descriptive text). 3) No general description of UTOPIA model is provided together with necessary references to previous work, where the model has been shown to be robust for the particular region under study.

More specific comments are: 1) The period 1961-1990 is hardly can be used to reflect "present climate". The period 1980-2010 is more appropriate. 2) p.3, line 30. There seem to be no physical reason for interpolating in time the precipitation and radiation fluxes by different methods. Does cubic spline interpolation conserves the sums of radiation fluxes? Were the output radiation data from RegCM3 presented as accumulated radiation sums or as fluxes? 3) p.4, line 1. "Short grasses are assumed to cover the whole domain". Not clear. Where there any other vegetation types in the domain? 4) The authors confined their analysis of soil moisture changes to examination of the water content of the top 5-cm-thick layer of the land model. Why not considering the whole root-occupied layer?

---

## Author Comment (AC1) · 27 Jan 2018

**Reply to the Comments by Referee #1 for Manuscript hess-2017-569**

We greatly appreciate the referee for thorough reading of the manuscript and valuable comments, which resulted in significant improvement of our manuscript. We have substantially revised the manuscript, following the referee's comments/suggestions. Below please find our item-by-item replies to the referee's comments.

**Major comments:**

- 1. The paper lacks any comparison with other papers. What is improvement in using the PRUDENCE output with respect, say, to other global and regional Earth system models? I guess, at least some comparison with the CMIP3/CMIP5 simulations should be included.
  - ⇒ We appreciate the referee for pointing this out. Although we have included some references related to this matter in the original manuscripts, we tried to include more relevant references in the revised manuscript, focusing on regional climate modeling over Europe and/or Italy. In this context, we added the following paragraphs by distributing to adequate positions in the revised manuscript (mostly appear just above Sec. 2.1), which include general overviews of CMIP3/CMIP5 and PRUDENCE, and summarized the previous studies that addressed the relevance of PRUDENCE in evaluating the regional climate and compared outputs from PRUDENCE and CMIP3:

In recent decades, the coupled atmosphere-ocean general circulation models (GCMs) improved significantly, and standard protocols of numerical climate model experiments were developed in the Coupled Model Intercomparison Project (CMIP) Phase 3 (CMIP3; Meehl et al., 2007); the CMIP3 dataset provided the scientific basis for the IPCC Fourth Assessment Report (AR4) based on the Special Report on Emissions Scenarios (SRES) emission scenario. The CMIP Phase 5 (CMIP5) dataset (Taylor et al., 2012) was developed based on the Representative Concentration Pathways (RCP) scenario that considers radiative forcing due to greenhouse gas concentration, and contributed to the IPCC Fifth Assessment Report (AR5).

Usually GCMs are calculated in relatively coarse grid spacings; thus representing the regional topography and climate inappropri-
ately (e.g., Bhaskaran et al., 2012). Therefore, downscaling of the GCM variables to regional scale is necessary for better depiction of regional climate: the dynamic downscaling uses regional climate models (RCMs) with a higher resolution (typically 10-50 km) and the same principles of dynamical and physical processes as GCMs (e.g., Wilby and Wigley, 1997; Christensen et al., 2007; Jury et al., 2015). It is demonstrated that RCMs significantly improves the model precipitation formulation (e.g., Frei et al., 2006; Gao et al., 2006; Buonomo et al., 2007; Boberg et al., 2009). In this context, an interdisciplinary project, called the Prediction of Regional Scenarios and Uncertainties for Defining European Climate Change Risks and Effects (PRUDENCE), had been undertaken aiming at providing high resolution climate change scenarios for Europe at the end of the 21st century via dynamical downscaling of global climate simulations (Christensen et al., 2007). Déqué et al. (2005), in comparison of results from GCMs and RCMs for the IPCC A2 radiative forcing, found that GCMs and RCMs behave similarly for the seasonal mean temperature with higher spread in GCMs; however, during summer, the spread of the RCMs - in particular in terms of precipitation — is larger than that of the GCMs, which indicates that the European summer climate is strongly controlled by parameterized physics and/or high-resolution processes. They also concluded that the PRUDENCE results were confident because the models had a similar response to the given radiative forcing.

In this study, we have employed RegCM3 from the PRUDENCE project to provide meteorological inputs to UTOPIA. Déqué et al. (2007) showed that, despite some uncertainties due to errors in sampling, model, and boundary and radiative forcings, the signal from the PRUDENCE ensemble is significant in terms of the mini-
mum expected 2 m temperature and precipitation responses for the IPCC A2 scenario. Jacob et al. (2007) demonstrated that RCMs in PRUDENCE generally reproduce the large-scale circulation of the driving GCM. Coppola and Giorgi (2010) assessed the 21st century climate change projections over Italy from the CMIP3 global and PRUDENCE regional model experiments, and found a broad agreement between the projections obtained with the CMIP3 and PRUDENCE ensembles. They also made a fine-scale (20 km) single model experiment using RegCM3, and found that the temperature biases of RegCM3 simulation are in line with those found for the individual PRUDENCE model simulations and both the temperature and precipitation changes through RegCM3 are in accordance with the CMIP3 and PRUDENCE results. These studies indicate that results from the PRUDENCE and CMIP3/CMIP5 experiments are roughly equivalent for the Mediterranean region and the Alpine sector.

- ⇒ We also included statements showing the consistency between our results with previous studies, specifically over the region of our study (i.e., Europe including the Alps and northern Italy), by referring to more relevant references. We actually added a separate subsection dedicated to this matter in Sec. 4 of the revised manuscript (see 4.4 Comparative discussion on previous works).
- 2. How specific are the UTOPIA simulations with respect to other land surface models? At least, some peculiarities of this model should be discussed in the context of other land surface models.
  - $\implies$  Although UTOPIA was shortly described in Section 2 of the original manuscript, we agree with the referee on this point. In the revised
manuscript, we have substantially amended this part by separating the original section "**2 Models and experimental setup**" into two independent sections as "**2 Description on models**" and "**3 Experimental design**"; then, in the updated Section 2, we included 2 subsections that are dedicated to RegCM3 and UTOPIA, respectively, by describing the main characteristics of the models in more detail.

- 3. Why the CMIP3 scenarios SRES A2/B2 are used? Now they are superceded by the RCP scenarios family.
  - ⇒ As the referee has remarked, the SRES scenarios are now superceded by the RCP scenarios; however, it does not necessarily mean that the SRES scenarios are useless or wrong. Furthermore, both scenario families are not completely different but have somewhat similar features. We included the following statements concerning about this issue adequately in the revised manuscript:

Riahi et al. (2011) mentioned that SRES A2 is comparable to RCP 8.5. Ward et al. (2011) found that the RCP 4.5 and SRES B1/A1T scenarios are broadly consistent with the fossil fuel production forecasts at that time. Rogelj et al. (2012; R12 hereafter) pointed out that the RCP scenarios span a large range of stabilization, mitigation and non-mitigation pathways, resulting in a larger range of temperature estimates than the SRES scenarios, which cover only non-mitigation scenarios (see Table 2 in R12); thus the SRES scenarios can be considered as a subset of the RCP scenarios in the context of global temperature projections by the end of the 21st century. They also indicated that the quantitative analysis of the differences between RCPs and SRES is hardly affected at all, and pairs with similar temperature projections over the 21st century can be found between the two sets: RCP 8.5 similar to A1FI, RCP6 HESSD
to B2, and RCP 4.5 to B1, respectively (see Table 3 in R12). Their findings ensure that SRES A2 is considered to be intermediate between RCP 6 and RCP 8.5. Matthews and Solomon (2013) showed that the cumulative CO2 emission and corresponding warming at near-term (2030) are approximately the same across all emission scenarios, whereas those at longer-terms (2100) are similar between close counterparts of the selected SRES and RCP scenarios: A1FI to RCP 8.5, A1B to RCP 6, and B1 to RCP 4.5, respectively. Baker and Huang (2014) found that the SRES A1B simulations in CMIP3 and the RCPs 4.5 and 8.5 simulations in CMIP5 produced common drying trend in the 21st century trend over the Mediterranean region. It is also indicated by Cabré et al. (2016) that SRES A2 has similarities to BCP 8.5 in terms of radiative forcing, future trajectories ( $\sim$ 8 W m-2 by 2100), and changes in global mean temperature (2.0 - 5.9°C for 2090-2099 compared to 1980-1999 for A2; 2.6 - 4.8°C for 2081-2100 compared to 1986-2005 for RCP 8.5). In R12, differences in warming rates existed between the scenario families due to different transient forcings; however, with a 30-year average for each scenario as in our study, the results and conclusions by using the SRES A2/B2 scenarios would not be significantly different from those by using the close RCP counterparts.

We have added statements addressing this point in the revised manuscript (see the early part of Sec. 3). Further discussions on this issue are provided in "Reply to Referee #2, Major comments #1" with citations of more relevant references.

4. And the most important one: Which new knowledge does the paper deliver to us?
 $\implies$  We admit that there exist several previous studies on the climate projections and related hydrologic changes around the Alps, using GCMs and/or RCMs; however, none of them studied projections of full water cycle by assessing all hydrologic components - precipitation, evapotranspiration, runoff and soil moisture — as in our study. Most of the previous studies focused on just some specific component(s) of water cycle, e.g., precipitation and/or surface runoff. For instance, Giorgi and Lionello (2008) studied climate change projections for the Mediterranean region, focusing on precipitation and temperature; Coppola et al. (2014) studied the impact of climate change on the Po basin, addressing discharge; and Torma et al. (2015) carried out ensemble RCM projections over the Alps, centering about precipitation. Compared to other previous studies, we think that our study is more exhaustive and has its own uniqueness: our study provides more complete analyses on all hydrologic components, including soil moisture, for both reference climate and future projections. Furthermore, with a companion paper on the land surface energy balance, we provide discussions on the linkages between the hydrologic and energy components. These enable us to better quantify some significant variations in the frame of changing climate in the Alpine area, in which the climatic change shows a larger variability. We have addressed these points adequately in the revised manuscript, which mostly appear in Sec. 4.4.

**Minor comments:**

- 1. *II. 6 and 7: I guess 'FC' in the abstract is difficult to understand. Please be more specific (e.g. replace it by time interval '2071–2100').*
  - ⇒ We agree with the referee and have modified the abstract as the referee suggested in the revised manuscript.
- 2. I. 10: 'Annual or seasonal variations...' What is the difference between them? Probably, authors mean 'interannual' in place of 'annual'?
  - ⇒ We appreciate the referee for pointing this out. To avoid any confusion, we modified the sentence by leaving just 'seasonal', since we were interested in underlining the variations within a year.

**References**

[revised manuscript text omitted]

---

## Author Comment (AC2) · 27 Jan 2018

**Reply to the Comments by Referee #2 for Manuscript hess-2017-569**

**General Comments:** The manuscript 'Climate change over the high-mountain versus plain areas: Effects on the land surface hydrologic budget in the Alpine area and northern Italy' presents climate change impacts on evapotranspiration, precipitation and soil moisture over the Alpine and northern Italy using regional climate model and land surface process model. Authors well deliver the changes of hydrological budget under climate change. However there are a few concerns described below.

We appreciate the valuable comments by the referee along with many helpful suggestions, which helped us improve the manuscript significantly. We have substantially revised the manuscript following the referee's comments/suggestions. In the following, we have provided an item-by-item reply to the referee's comments.

**Major comments:**

- 1. Authors used RegCM3 in this study. Newer version model dose not necessary mean that having better performance but authors need to justify why older version model with older scenario (AR4) was employed in this study.
  - ⇒ We admit the existence of a newer version of RegCM (i.e., RegCM4) and totally agree with the referee that newer version model does not necessarily has better performance. We decided to employ RegCM3 for the following reasons:
    - RegCM3 had been employed in several important projects, including PRUDENCE, ENSEMBLES and CECILIA, whose outputs had been used in numerous studies focusing on Europe (e.g., Christensen and Christensen, 2007; Blenkinsop and Fowler, 2007; Ballester et al., 2010; Coppola and Giorgi, 2010; Herrera et al., 2010; Rauscher et al., 2010; Kyselý et al., 2011; Torma et al., 2011; Heinrich et al., 2014; Skalák et al., 2014; Faggian, 2015);
    - It had also been widely used, even most recently, for the studies of climate projections or sensitivities/evaluations over a geographical region including the target areas in our study — the Alpine and adjacent areas (e.g., Gao et al., 2006; Smiatek et
al., 2009; Coppola and Giorgi, 2010; Im et al., 2010; Coppola et al., 2014; Nadeem and Formayer, 2016; Alo and Anagnostou, 2017);

3) Since a plenty of model outputs were available from several relevant projects (e.g., PRUDENCE, ENSEMBLES, CECILIA, etc.) and we had limited computing resources and man power for exploring all available data sources, we decided to select a wellknown model which had been extensively used for such kind of studies.

These points are now clearly addressed in the revised manuscript (see the early part of Sec. 2).

⇒ We also acknowledge that the scenarios used here (SRES A2/B2) are older than the RCP scenarios used in IPCC AR5. However, scenarios are designed to depict possible future developments with some uncertainties that reflect different understandings of the current/intermediate socio-economic and/or greenhouse gas emission circumstances; thus, each scenario has its own value and philosophy, and older scenarios do not necessarily mean that they are useless or wrong. Numerous previous studies on climate projections/impacts had been conducted based on the SRES scenarios, which we cannot totally neglect even though new scenarios have emerged. Rather, there have been some studies to check similarities/differences between the two scenario sets for a given projection period or to address the value of using both scenario sets for future climate projections. We added the following statements in appropriate positions in the revised manuscript. In addition, explanations on more relevant references are provided in "Reply to Referee 1" (see Major comments #3 and references therein).

In the last decade, numerous studies on climate projec-

**HESSD**
tions/impacts had been conducted using the SRES scenarios. After the emergence of the new RCP scenarios, there have been studies to check similarities/differences between the two scenario sets for a given projection period (e.g., Rogelj et al., 2012; Baker and Huang, 2014) or to address the value of using both scenario sets for future climate projections (e.g., Peters et al., 2013; O'Sullivan et al., 2016; Nolan et al., 2017). It turns out that both SRES and RCP scenarios are generally in good agreements, for pairs of closest counterparts, in projecting climate in the 21st century. For example, Rogeli et al. (2012) pointed out that the RCP scenarios spanned a larger range of temperature estimates than the SRES scenarios, and indicated similar temperature projections for pairs between the two scenario sets: RCP 8.5 similar to A1FI, RCP6 to B2, and RCP 4.5 to B1, respectively. Baker and Huang (2014) reported a common drying trend, over the Mediterranean region, between the CMIP3 simulations based on SRES A1B and the CMIP5 simulations based on RCPs 4.5 and 8.5. Peters et al. (2013) projected global warming through all available emission scenarios, showing that RCP 8.5 and SRES A1FI and A2 lead to the highest temperature projections and RCP3-PD (peak and decline in concentration) would keep global warming below 2°C in 2100. Most recently, O'Sullivan et al. (2016) and Nolan et al. (2017) assessed impacts of climate change on temperature and rainfall, respectively, by mid-21st century in Ireland using both SRES and RCP scenarios, and provided a wide range of possible climate projections. O'Sullivan et al. (2016) found that future summers had the largest projected warming under RCP 8.5 while future winters had the greatest warming under A1B and A2. Nolan et al. (2017) created a medium-to-low emission ensemble using the RCP4.5 and B1 scenario simulations and a high
emission ensemble using the RCP8.5, A1B and A2 simulations, which enabled to have 25 high and 21 medium-to-low emission ensemble comparisons: they found significant projected decreases in mean annual, spring and summer precipitation amounts — largest for summer, with different reduction range for different scenario ensemble.

Furthermore, the SRES scenarios have often been adopted in most recent studies even long after the release of the RCP scenarios because the old scenarios were in accord with their objectives (e.g., Dunford et al., 2015; Jaczewski et al., 2015; Kiguchi et al., 2015; Kim et al., 2015; Casajus et al., 2016; Harrison et al., 2016; Mamoon et al., 2016; Stevanović et al., 2016; Tukimat and Alias, 2016; Zheng et al., 2016; Hassan et al., 2017; Park et al., 2017; da Silva et al., 2017). We employed the SRES marker scenarios because of their long-term consistency in assessing the impact of climate change on global/regional factors of socio-economy and environment during the last decade — including air quality (Jacob and Winner, 2009; Carvalho et al., 2010), water guality/resources (Wilby et al., 2006; Shen et al., 2008, 2014; Luo et al., 2013), energy (Hoogwijk et al., 2005; van Vliet et al., 2012), agriculture/forestry (Lavalle et al., 2009; Calzadilla et al., 2013; Stevanović et al., 2016; Zubizarreta-Gerendiain et al., 2016), fisheries (Barange et al., 2014; Lam et al., 2016), health/disease (Patz et al., 2005; Giorgi and Diffenbaugh, 2008; Ogden et al., 2014), climate/weather extremes (Déqué, 2007; Marengo et al., 2009; Jiang et al., 2012; Rummukainen, 2012), wildfire (Liu et al., 2010; Westerling et al., 2011), ecosystem/biodiversity (Araújo et al., 2008; Feehan et al., 2009; Jones et al., 2009; Fronzek et al., 2012; Walz et al., 2014), and so forth. Although an ensemble approach with all possible sce-
narios would increase the spread of hydrologic budget simulations, due to the limited resources, we decided to select two representative marker scenarios: A2 as the higher-end and B2 as the lowerend emission scenario, respectively.

All these points addressed above are adequately reflected in the revised manuscript (see the early part of Sec. 3).

- 2. In this study, authors employed single RCM and single land model. Authors need to discuss about model uncertainties comparing to the multi-model approaches.
  - $\implies$  We totally accept that a single model approach has relatively larger uncertainty: it is desirable to employ an ensemble approach, using multiple models and/or initial conditions, to estimate the range of climate projections. Our decision to employ the single-model approach is mainly due to limitation in computing resources and man power to perform multi-model ensemble simulations for both RCM and LSM. Given such limitations, a high-resolution single model is often not a bad choice, compared to an ensemble of coarse multi-models, especially over a complex terrain. Coppola and Giorgi (2010) pointed out that the CMIP3 GCMs showed a much larger range of bias for temperature and precipitation than the PRUDENCE RCMs: they also made a fine-scale (20 km) single model experiment using RegCM3 and found that both the temperature and precipitation changes through RegCM3 were in line with the CMIP3 and PRUDENCE ensemble results. Generally speaking, multi-model ensembles tend to decrease the errors compared to an individual model; however, due to the averaging operation (e.g., ensemble mean), the spatial and temporal variability of the signal tends to decrease. Furthermore, many previous studies on various climate change impacts/projections had been performed using the single RCM approach (e.g., Dankers and Feyen, 2008; Beniston, 2009; Im et al. 2010; Krüger et al., 2012; Zanis et
al., 2012; Tainio et al., 2013; Park et al., 2017). However, as uncertainty of the projected changes related to model bias and ensemble variability is quite large, future projections based on a single RCM should be interpreted with caution. Further research is needed to obtain more robust results from an ensemble approach. In the revised manuscript, we have addressed these points and mentioned limitations of our study in terms of the single-model approach (see the last part of Sec. 3).

- 3. I do believe there are quite a few previous studies over the study region. Authors need to introduce them.
  - ⇒ Following the referee's suggestion, we have added statements in the revised manuscript by citing more relevant references of the previous studies on climate projections over the study region (generally Europe including the Alps and northern Italy). We actually added a separate subsection dedicated to this matter in Sec. 4 of the revised manuscript (see 4.4 Comparative discussion on previous works).
- 4. Authors discussed about possible agricultural impact due to the lower soil moisture. However, in this study, vegetation type in UTOPIA were set single type of vegetation (short grasses) all over the domain. Have you done any sensitivity test on vegetation types?
  - ⇒ We decided to set the vegetation type equal for all grid points (i.e., short grasses) for the following reasons: 1) for the "reference climate", to avoid any problem in interpretation of results due to the differences in vegetation; and 2) for the "future climate", to alleviate the uncertainty in vegetation type at the end of 21st century. In terms of meteorological variables, this is not a bad assumption because most observation stations are normally installed over short grasses.

HESSD
By the way, in terms of plant height, root depth and vegetation characteristics, short grasses can be roughly regarded as most common cereals (wheat, maize, etc.), and would not be quite different from such kind of agricultural products. Finally, we have also performed simulations using the "true" vegetation (as deduced by detailed databases), and the results with the pastures and agricultural areas have generally been confirmed, though the numerical values of the variables were slightly different. Unfortunately, we did not publish papers about this topic yet. We have addressed these points in the revised manuscript (see the middle part of Sec. 3; staring last paragraph of page 8).

**Minor comments:**

- 1. Page 1. Remove all acronyms in abstract.
  - ⇒ We minimized the use of acronyms in Abstract; however, we kept acronyms of the model names (i.e., RegCM3 and UTOPIA) because most readers recognize the model names by their acronyms.
- 2. Page 2, line 15. Global circulation models. Is this different one as GCMs previously defined?
  - ⇒ We appreciate the reviewer pointing this out. Usually GCM can represent either Global Climate Model or General Circulation Model. As the global climate is usually simulated using the general circulation model, we re-defined GCM as Global Circulation Model and modified the text accordingly in the revised manuscript.
- 3. Page 3, line 22-24. Related with major comments 1. EURO-CORDEX has RegCM4 with higher resolution.
- ⇒ We have explained the reason to choose RegCM3 in Major Comments #1. However, as the referee pointed out, the statement "... one of the existing datasets with the highest resolution currently available (20 km)" may not be true because of the existence of RegCM4 with higher resolution (about 12 km). We have modified the expression in the revised manuscript as "... one of the high-resolution datasets currently available".
- 4. Page 4, line 5-19. Move to methods section.

 $\implies$  Done.

- 5. Page 4, line 20. Energy variables are critical part of your argument. Please include the figures as a supplementary.
  - ⇒ The land surface energy balance is an important issue itself, and we prepared a companion paper dealing with the energy budget components in the same study area — it is ready to be submitted. We put the companion paper to the References and cited appropriately in the revised manuscript.
- 6. Page 4, line 25. I cannot find the PR minimum shifts in the figure.
  - $\implies$  Actually the figure shows the trends of variables for PC in Fig. 2a, and the anomalies of variables (i.e., the differences between FCs and PC) in Figs. 2b and c. Thus, it is quite difficult to see the trends of variables in FCs from Figs. 2b and c. By looking at the individual trends related to A2 and B2 simulations (not shown in the paper), the mentioned shift becomes noticeable. To avoid any confusion, we have modified the expression "with the PR minimum shifted to August in FCA2" to "with the PR minimum shifted to August in FCA2" to "with the PR minimum shifted to August in FCA2".
- 7. Page 4, line 26. ET of  $FC_{B2}$  shows double peaks rather than shifting in Fig. 2b.
- $\implies$  As we explained above in #6, it is because the lines referred to FCB2 represent the anomalies, i.e., the differences between FCB2 and PC (i.e., FCB2 minus PC). Again we modified the expression "*the ET maxima shift towards July/August, in both FC*A2 and FCB2" to "*the ET maxima shift totowards July/August, in both FC*A2 and FCB2" to "*the ET maxima shift to-*
- 8. Page 4, line 29-31. It looks like the large variation stems on future PR variation. Can you explain why PR has large variation?
  - ⇒ As reported by some other studies (e.g., Gao et al., 2006), in winter the increase in southwesterly flow across the Alps causes a maximum of positive precipitation change over the southern Alps while in autumn the main circulation change is in the easterly and southeasterly direction. This explains the positive precipitation change at south of the Alps. We have added this explanation in the revised manuscript.
- 9. Page 6 line 27-29. Can you include the names of geographical location on the map (e.g. Fig. 1)
  - $\implies$  We added additional map representing the names of geographical locations in Figure 1, as requested by the reviewer.

**References**

[revised manuscript text omitted]

**HESSD**

---

## Author Comment (AC3) · 27 Jan 2018

**Reply to the Comments by Referee #3 for Manuscript hess-2017-569**

*General Comments: The paper presents assessment of changes of land water budget terms in Northern Italy under future climate changes. The regional climate model RegCM3 simulations are used as a forcing for the land surface scheme UTOPIA. The modeled seasonal and spatial patterns of precipitation, evapotranspiration, runoff, soil storage, net radiation are examined, and implications for regional economies are formulated.*

[Figure]

$\implies$ We appreciate the referee for careful reading and valuable comments, which helped us improve the manuscript significantly. We have revised the manuscript substantially, following the referee's comments/suggestions. Please find our item-by-item responses to the referee's comments below.

***Major comments:***

1. *There is no proper comparison of results obtained to other similar studies conducted for this region, elucidating what is the new knowledge attained. Some of other relevant papers are cited (Lautenschlager et al., 2008; Jacob et al., 2007), but the comparison is very limited.*

   $\implies$ We have included statements showing the consistency between our results with previous studies, specifically over the region of our study (i.e., generally Europe including the Alps and northern Italy), by referring to more relevant references. We actually added a separate subsection dedicated to this matter in Sec. 4 of the revised manuscript (see **4.4 Comparative discussion on previous works**).

   $\implies$ In terms of the new knowledge attained, we also have replied to the other referee's comment (Major comments #4 by Referee #1), and it is repeated here:

   We admit that there exist several previous studies on the climate projections and related hydrologic changes around the Alps, using GCMs and/or RCMs; however, none of them studied projections of full water cycle by assessing all hydrologic components — precipitation, evapotranspiration, runoff and soil moisture — as in our study. Most of the previous

studies focused on just some specific component(s) of water cycle, e.g., precipitation and/or surface runoff. For instance, Giorgi and Lionello (2008) studied climate change projections for the Mediterranean region, focusing on precipitation and temperature; Coppola et al. (2014) studied the impact of climate change on the Po basin, addressing discharge; and Torma et al. (2015) carried out ensemble RCM projections over the Alps, centering about precipitation. Compared to other previous studies, we think that our study is more exhaustive and has its own uniqueness: our study provides more complete analyses on all hydrologic components, including soil moisture, for both reference climate and future projections. Furthermore, with a companion paper on the land surface energy balance, we provide discussions on the linkages between the hydrologic and energy components. These enable us to better quantify some significant variations in the frame of changing climate in the Alpine area, in which the climatic change shows a larger variability. We have addressed these points adequately in the revised manuscript, which mostly appear in Sec. 4.4.

2. *The physical analysis of simulation results is somewhat superficial. The simple effects are explained, whereas the more complicated ones (like the absence of spatial correlation between evapotranspiration and precipitation, lines 1-4 on p.7) are commented by too general statements. In this respect, the striking separation of regions with large dry and wet days numbers anomalies at Figure 7 is left without deserving physical analysis (lines 25+, p.7 are merely descriptive text).*

$\implies$ We appreciate the referee for pointing this out. In the revised manuscript, we have tried to include more physical interpretations in our results. For example, for Fig. 7, we may extend the analyses and interpretations in previous figures: Overall, in the plain areas including the Po Valley, $\Delta$ET is positive while $\Delta$PR is weekly negative and $\Delta$SM is moderately negative (especially

during summer as in Figs. 2 and 3). With more significant overall increases in NR over plains, the combined effect will bring about larger evaporation and lower soil moisture, thus overall increase in the number of dry days, mostly attributed to much drier climate in summer. Meanwhile, over the high-mountain areas, PR, SR and SM increase while ET shows little variation in spring and winter (see Figs. 4 and 5). As SM is large over high mountains, we have more source of atmospheric moisture through evaporation there. Then, through the combined effect of terrain-induced convective motion, increase in NR (though less significant) and pre-existing snow, we can have more snow melting (during spring) and more liquid precipitation (especially during winter), resulting in more wet days, again mostly attributed to much wetter climate in winter. Such kind of discussions with physical interpretations are appropriately added in the revised manuscript.

3. *No general description of UTOPIA model is provided together with necessary references to previous work, where the model has been shown to be robust for the particular region under study.*

$\implies$ We appreciate the referee for pointing this out. Although UTOPIA was shortly described in Section 2 of the original manuscript, we agree with the referee on this point. In the revised manuscript, we have substantially amended this part by separating the original section "**2 Models and experimental setup**" into two independent sections as "**2 Description on models**" and "**3 Experimental design**"; then, in the updated Section 2, we included 2 subsections that are dedicated to RegCM3 and UTOPIA, respectively, by describing the main characteristics of the models in more detail. We have also added a paragraph that cite relevant references to previous work, where UTOPIA has demonstrated its robustness for the region of our study.

**Specific comments:**

1. *The period 1961-1990 is hardly can be used to reflect "present climate". The period 1980-2010 is more appropriate.*

    ⟹ We generally agree with the referee about this point, and it is common nowadays that the climatological 30-year statistics are updated every ten years. However, the former period 1961–1990 still remains the official normal period defined by WMO, and numerous previous studies on climate change projections/impacts, including several projects (e.g., CMIP3/CMIP3, PRUDENCE, ENSEMBLES and CECILIA), employed this period as "present climate" (or control/reference/baseline period), even most recently (e.g., to mention just a few, Giorgi and Lionello, 2008; Smiatek et al., 2009; Ciscar et al., 2011; Kyselý et al., 2011; Torma et al., 2011; Heinrich et al., 2014; Perez et al., 2014; Skalák et al., 2014; Belda et al., 2015; Dunford et al., 2015; Faggian, 2015; Casajus et al., 2016; Harrison et al., 2016; Gang et al., 2017; Paeth et al., 2017). Furthermore, as requested by the referees, we need to make comparisons between our results and previous studies over the region of current study. For this purpose and fair comparisons, we need to keep consistency with the period that represent "present climate" (i.e., 1961–1990) in many previous studies. On the other hand, we agree with the referee that this period may not reflect "present climate" in practical sense; thus we decided to define it as "reference climate", which can be acceptable in general sense. We have modified "present climate (PC)" to "reference climate (RC)" in the text and figures in the revised manuscript. This issue is now addressed at the beginning of Sec. 3.

2. *p.3, line 30. There seem to be no physical reason for interpolating in time the precipitation and radiation fluxes by different methods. Does cubic spline interpolation conserves the sums of radiation fluxes? Were the output radiation data from RegCM3 presented as accumulated radiation sums or as fluxes?*

$\implies$ We applied the cubic spline to the non-intermittent variables like temperature, humidity, and radiation (flux), whereas we simply redistributed the intermittent variable, e.g., precipitation to keep its sum. There is a reason for having used different methods for radiation and precipitation: the input data of precipitation was the precipitation cumulated over the timesteps of the RCM output, and this datum cannot be interpolated with splines. Of course, we could have converted precipitation to precipitation rates, interpolated them using splines, and then reconverted to cumulated precipitations over the smaller timestep of UTOPIA. However, the result of such a complicated procedure was almost equivalent to using the method described in the text. Regarding radiation, we used the splines for the sake of uniformity with other variables (wind components were also interpolated in this way). We further controlled some unrealistic values (e.g., negative radiations): we controlled the daily means (or cumulated values) of input data (from RegCM3) and output data (for UTOPIA) from the spline interpolation method to be equivalent, with positive or null values. We have addressed these points in the revised manuscript.

3. *p.4, line 1. "Short grasses are assumed to cover the whole domain". Not clear. Where there any other vegetation types in the domain?*

$\implies$ The domain includes the Alps, the Apennines, off-alpine and hilly areas, and plains; thus there is a wide range of vegetation in the domain. Regarding plains and hilly areas, vegetation includes pastures, grasslands and some forested areas: mountain areas are mostly covered by trees, and the highest parts are without vegetation or covered by permanent ice (few grid points). We decided to set the vegetation type equal for all grid points (i.e., short grasses) for the following reasons: 1) for the "reference climate", to avoid any problem in interpretation of results due to the differences in vegetation; and 2) for the "future climate", to alleviate the uncertainty in vegetation type

at the end of 21st century. In terms of meteorological variables, this is not a bad assumption because most observation stations are normally installed over short grasses.

By the way, in terms of plant height, root depth and vegetation characteristics, short grasses can be roughly regarded as most common cereals (wheat, maize, etc.), and would not be quite different from such kind of agricultural products. Finally, we have also performed simulations using the "true" vegetation (as deduced by detailed databases), and the results with the pastures and agricultural areas have generally been confirmed, though the numerical values of the variables were slightly different. Unfortunately, we did not publish papers about this topic yet. We have addressed these points in the revised manuscript.

4. *The authors confined their analysis of soil moisture changes to examination of the water content of the top 5-cm-thick layer of the land model. Why not considering the whole root-occupied layer?*

$\implies$ Actually, for the short grass vegetation category considered in these simulations, the root layer is only 5 cm deep, as the grass is only 10 cm high. Despite this value seems too low, it represents the typical height for the landscapes of Italian Po valley (at least in its portion occupied by natural vegetation). Furthermore, the upper soil layer represents the greatest effect of the atmosphere-land surface-soil interactions. Given that we are interested in the present vs. future hydrologic budget components, we decided to focus on the top soil layer. More specifically, we wanted to show the water content of the soil layer that represents the largest variations of moisture: it is subjected to direct evaporation, to the transpiration from vegetation roots, to the gravitational drainage to the second soil layer, to the capillary suck of moisture from the second soil layer, and finally to the eventual precipitation, eventual vegetation drainage, and eventual snow runoff. In other occasions,

we have also analyzed the behavior of the full root zone layer, and/or of a deeper portion of soil; however, we noticed that the behavior of the upper portion of soil can also give a qualitative and quantitative idea of what is happening in the deeper soil. Last but not least, if we consider deeper portions of soil, the behavior can differ depending on the soil property such as hydraulic conductivity: soil with a large clay component creates a larger vertical moisture gradient than that with a large sand component. We have addressed these points in the revised manuscript.

**References**

Belda, M., Skalák, P., Farda, A., Halenka, T., Déqué, M., Csima, G., Bartholy, J., Torma, C., Boroneant, C., Caian, M., and Spiridonov, V.: CECILIA regional climate simulations for future climate: Analysis of climate change signal, Adv. Meteorol., 2015, doi: 10.1155/2015/354727, 2015.

Casajus, N., Périé, C., Logan, T., Lambert, M.-C., de Blois, S., Berteaux, D.: An objective approach to select climate scenarios when projecting species distribution under climate change, PLoS ONE, 11, e0152495, doi: 10.1371/journal.pone.0152495, 2016.

Ciscar, J.-C., Iglesias, A., Feyen, L., Szabó, L., Van Regemorter, D., Amelung, B., Nicholls, R., Watkiss, P., Christensen, O. B., Dankers, R., Garrote, L., Goodess, C. M., Hunt, A., Moreno, A., Richards, J., and Soria, A.: Physical and economic consequences of climate change in Europe, Proc. Nat. Acad. Sci. USA, 108, 2678–2683, doi: 10.1073/pnas.1011612108, 2011.

Dunford, R. W., Smith, A. C., Harrison, P. A., Hanganu, D.: Ecosystem service provision in a changing Europe: adapting to the impacts of combined climate and socio-economic change, Landscape Ecol., 30, 443–461, doi: 10.1007/s10980-014-0148-2, 2015.

Faggian, P.: Climate change projections for Mediterranean region with focus over Alpine region and Italy, J. Environ. Sci. Eng. B, 4, 482–500, doi: 10.17265/2162-5263/2015.09.004, 2015.

Gang, C., Zhang, Y., Wang, Z., Chen, Y., Yang, Y., Li, J., Cheng, J., Qi, J., and Odeh, I.: Modeling the dynamics of distribution, extent, and NPP of global terrestrial ecosystems in response to future climate change, Global Planet. Change, 148, 153–165, 2017.

Giorgi, F., and Lionello, P.: Climate change projections for the Mediterranean region, Global Planet. Change, 63, 90–104, doi: 10.1016/j.gloplacha.2007.09.005, 2008.

Harrison, P. A., Dunford, R. W., Holman, I. P., and Rounsevell, M. D. A.: Climate change impact modelling needs to include cross-sectoral interactions, Nature Clim. Change, 6, 885–890, doi: 10.1038/nclimate3039, 2016.

Heinrich, G., Gobiet, A., and Mendlik, T.: Extended regional climate model projections for Europe until the mid-twentyfirst century: combining ENSEMBLES and CMIP3, Clim. Dyn., 42, 521–535, doi: 10.1007/s00382-013-1840-7, 2014.

Kyselý, J., Gaál, L., Beranová, R., Plavcová, E.: Climate change scenarios of precipitation extremes in Central Europe from ENSEMBLES regional climate models, Theor. Appl. Climatol., 104, 529–542, doi: 10.1007/s00704-010-0362-z, 2011.

Paeth, H., Vogt, G., Paxian, A., Hertig, E., Seubert, S., Jacobeit, J.: Quantifying the evidence of climate change in the light of uncertainty exemplified by the Mediterranean hot spot region, Global Planet. Change, 151, 144–151, 2017.

Perez, J., Menendez, M., Mendez, F. J., Losada, I. J.: Evaluating the performance of CMIP3 and CMIP5 global climate models over the north-east Atlantic region, Clim. Dyn., 43, 2663–2680, doi: 10.1007/s00382-014-2078-8, 2014.

Skalák, P., Déqué, M., Belda, M., Farda, A., Halenka, T., Csima, G., Bartholy, J., Caian, M., and Spiridonov, V.: CECILIA regional climate simulations for the present climate: validation and inter-comparison, Clim. Res., 60, 1–12, doi: 10.3354/cr01207, 2014.

Smiatek, G., Kunstmann, H., Knoche, R., and Marx, A.: Precipitation and temperature statistics in high-resolution regional climate models: Evaluation for the European Alps, J. Geophys. Res., 114, D19107, doi: 10.1029/2008JD011353, 2009.

Torma, C., Coppola, E., Giorgi, F., Bartholy, J., and Pongrácz, R.: Validation of a high-resolution version of the regional climate model RegCM3 over the Carpathian basin, J. Hydrometeorol., 12, 84–100, doi: 10.1175/2010JHM1234.1, 2011.

---

## Editor Decision (ED1)

I appreciate and thank the authors for their clear and concise response to feedback from all reviewers.  The authors' responses were thoughtful and thorough.  The reviewers find that the revised version of the manuscript satisfactory addresses their comments and can be published as is. I agree with the reviewers' opinion and recommend the manuscript for publication in HESS.

Alexander Gelfan,
Handling Editor

| Principal Criteria | Excellent (1) | Good (2) | Fair (3) | Poor (4) |
|---|---|---|---|---|
| **Scientific Significance:** Does the manuscript represent a substantial contribution to scientific progress within the scope of *Hydrology and Earth System Sciences* (substantial new concepts, ideas, methods, or data)? | | + | | |
| **Scientific Quality:** Are the scientific approach and applied methods valid? Are the results discussed in an appropriate and balanced way (consideration of related work, including appropriate references)? | | + | | |
| **Presentation Quality:** Are the scientific results and conclusions presented in a clear, concise, and well-structured way (number and quality of figures/tables, appropriate use of English language)? | | + | | |